# Mosquito salivary sialokinin reduces monocyte activation and chikungunya virus-induced inflammation via neurokinin receptors

Siew-Wai Fong [1] ✉, Jeslin J. L. Tan[2], Vaishnavi Sridhar [3], Siti Naqiah Amrun[1], Vanessa K. X. Neo[1], Nathan Wong[1], Bernett Lee [4], Yi-Hao Chan [1], Anthony Torres-Ruesta [1], Liang Hui. Loo [1], Anna X. Y. Loo[1], Sarah K. W. Tan[1], Rhonda S. L. Chee[1], Tze-Kwang Chua [5], Angeline Rouers[1], Guillaume Carissimo [1,4,6], Fok-Moon Lum[1], Yee-Sin Leo [4,7,8], Laurent Renia[1,4,9], R. Manjunatha Kini [3,10,11] & Lisa F. P. Ng [1,4,12] ✉

Global warming is expanding mosquito habitats and increasing mosquito-borne diseases. In tropical and sub-tropical regions, chikungunya virus (CHIKV) transmitted by Aedes mosquitoes has become a major concern due to the debilitating chronic joint disease it causes. Mosquito saliva contains bioactive factors that enhance viral infection, with sialokinin identified as a key contributor to vascular leakage and viral spread in mice. Here, we demonstrate that sialokinin binds to neurokinin receptors and restricts the activation of human myeloid cells. Mechanistically, sialokinin facilitates early viral dissemination, as evidenced by increased viral load in the contralateral footpad at 1 day post-infection, and significantly reduces circulating CD169+ monocytes while suppressing IFN-γ-producing T-cell-driven inflammation, as reflected by reduced joint footpad swelling in female CHIKV-infected mice. Clinically, patients with severe CHIKV disease exhibited higher levels of IgG antibodies against sialokinin, which correlated with higher viral loads and systemic inflammatory markers. Our findings highlight the multifaceted role of sialokinin in facilitating early viral dissemination and modulating host immunity during CHIKV infection. Given the growing threat of mosquito-borne diseases in a warming, disease-burdened world, targeting mosquito salivary factors like sialokinin could offer a novel therapeutic strategy to mitigate viral-induced inflammation and improve clinical outcomes.

Mosquito-borne diseases like malaria, dengue, Zika, and chikungunya are major global health threats that will likely increase as climate change intensifies[1–3]. Warmer temperatures and shifting weather patterns are expanding the geographical range of mosquito vectors[4,5], enabling them to thrive in new regions[6] and intensifying the spread of the diseases they carry. While current control measures, including vaccines, insecticides, and antimicrobial therapies, have made strides, their effectiveness is increasingly compromised by antimicrobial

resistance[7,8], insecticide resistance or tolerance[9]. These challenges underscore the urgent need for alternative strategies to control transmission and mitigate the growing threat of mosquito-borne diseases in a warming world.

One promising avenue for intervention focuses on targeting mosquito salivary factors, which play a key role in facilitating pathogen transmission. When infected mosquitoes bite, they inject both the pathogen and their saliva into the human host. This saliva contains numerous pharmacologically active components that facilitate blood-feeding by inhibiting coagulation[10] and platelet aggregation[11,12], suppressing thrombin-like activities[13], and dilating host blood vessels[14]. Notably, mosquito saliva has profound immunomodulatory effects on the human host, with immune responses to mosquito bites detectable up to seven days post-exposure[15].

Recent studies have indicated that mosquito saliva can enhance the infectivity of pathogens by modulating host's immune responses[16,17]. Chikungunya virus (CHIKV) is a mosquito-transmitted alphavirus that has, in recent years, emerged in many new locations globally, including South America and Southeast Asia[18,19]. CHIKV is primarily transmitted by infected female mosquitoes[20] of the *Aedes aegypti* and *A. albopictus* species[21]. The virus may have evolved into a more severe form of disease in the past few decades[22]. CHIKV infection can cause fever and joint pain. In some cases, it can lead to neurological complications[23] and tends to be more severe in infants[24]. Until now, treatments remain mainly symptomatic[25].

Targeting specific immune modulators in mosquito saliva presents a new frontier for therapeutic development. For instance, sialokinin, a vasodilatory peptide abundantly expressed in female *Aedes aegypti* saliva, contains the Phe-Xaa-Gly-Leu-Met-NH₂ carboxyl-terminal sequence required for binding to tachykinin-specific receptors[26]. Human neurokinin (NK) receptors are a family of tachykinin-specific receptors involved in neurotransmission and inflammatory responses[27]. They exist in three variants (NK₁R, NK₂R, and NK₃R) and are widely expressed in tissues such as the nervous system[28], immune cells[29], and smooth muscle[30], with NK₃R being primarily conserved in the central nervous system[31]. Previous studies have shown that sialokinin contributes to increased infectivity of Semliki Forest virus, an alphavirus closely related to CHIKV, by disrupting endothelial barrier integrity[32]. Here, we demonstrate the immunomodulatory effect of sialokinin on human myeloid cells and its impact on CHIKV immunopathology in vivo. We also report an antibody response to sialokinin and its association with disease progression in patients infected with CHIKV. Our findings suggest sialokinin as a possible therapeutic target for modulating immune responses and mitigating the inflammation associated with mosquito-borne viral infections.

## Results

### Aedes aegypti sialokinin binds to human NK receptors

Sialokinins are present as two isopeptides (sialokinin I and II) in the mosquito saliva, with Asn1 deamidated to Asp1, the only difference between them (Supplementary Fig. 1a). Due to their high sequence similarity to human tachykinins, notably in the carboxyl-terminal region (Supplementary Fig. 1a), both sialokinins might have the ability to bind to human NK receptors.

We assessed the binding affinity of sialokinin I, the predominant form in *Aedes* saliva, for human NK receptors using a calcium flux assay. Sialokinin I induced a dose-dependent increase in intracellular calcium ion influx in Chinese hamster ovary (CHO)-K1 cells expressing human NK₁R, NK₂R, or NK₃R (Supplementary Fig. 1b), acting as a full agonist for all three receptors. Analysis of the concentration dependence of the peak response to sialokinin showed EC₅₀ values of $0.052 \pm 0.028$ nM for the NK₁R, $0.288 \pm 0.107$ nM for the NK₂R, and $1.289 \pm 0.281$ nM for the NK₃R (Supplementary Fig. 1b). No response was observed in non-transfected CHO-K1 cells (Supplementary Fig. 1b).

These findings indicate that sialokinin I can bind functionally to human NK receptors in vitro.

We also verified the specificity of NK₁R antagonist CP-96345 and NK₂R antagonist GR-159897 using calcium flux assays. Cells were pre-incubated with varying concentrations of the antagonists before stimulation with sialokinin. Pre-incubation with CP-96345 shifted the EC₅₀ of sialokinin in NK₁R-expressing CHO cells, without affecting the E_max value (Supplementary Fig. 1c). GR-159897 also inhibited sialokinin binding to NK₂R. These results confirm that CP-96345 and GR-159897 specifically target NK₁R and NK₂R, respectively (Supplementary Fig. 1c).

### Sialokinin reduces the activation of human monocytes and macrophages

We next examined whether the sialokinin peptides could regulate the activation status of human monocytes, the primary immune cells that express NK₁R and NK₂R[33]. Primary monocytes were isolated from human whole blood and stimulated with a 10 µM concentration of sialokinin, which is physiologically relevant[26,34,35] to the amount typically introduced during mosquito bites. Using Oxford Nanopore Sequencing, we produced an RNA-Seq dataset of primary human monocytes from three biological replicates. On average, 2 million reads were obtained per sample, and 55% of reads were mapped to the human reference genome. Data from RNA-Seq were pre-processed according to the EPI2ME analysis pipeline[36]. We identified a total of 12 differentially expressed genes (DEGs) with false discovery rate (FDR) < 0.05 between sialokinin-treated vs. non-treated primary human monocytes, of which five genes (*RAC2*, *ALOX5AP*, *CXCL5*, *FAM120A*, and *SPP1*) were upregulated and seven genes (*SIGLEC1, IFIT2, IFI44L, TPP1, BEST, ACP5,* and *HLA-DMB*) were downregulated in response to sialokinin treatment (Fig. 1a). Sialokinin limited monocyte activation, as evidenced by the downregulation of several genes involved in type 1 interferon (IFN) signaling (*SIGLEC1, IFIT2, IFI44L*, and *ACP5*) and antigen presentation (*HLA-DMB*).

We further assessed sialokinin's effects on human monocyte activation by fluorescence-activated cell sorting (FACS) analysis. We treated human monocytes with increasing concentrations of sialokinin (0.1, 1, and 10 µM). Monocytes were stained with antibodies against the cell surface markers CD45, CD14, CD16, CCR2, HLA-DR, CD11b, and CD169, and monocyte subsets were defined based on their expression of these markers. Sialokinin-treated monocytes showed significantly decreased levels of the activation markers HLA-DR, CD16, and CD169 on their surface, with CD169 having the most significant reduction (Supplementary Fig. 2a and Fig. 1b) at 10 µM sialokinin. A similar effect was observed in monocyte-derived macrophages (MDMs) (Supplementary Fig. 2a). These effects of downregulation were concentration-dependent and were consistent across donors in both sets of monocytes (Fig. 1b) and MDMs (Supplementary Fig. 2a). Additionally, no decrease in monocyte activation was observed when cells were treated with scrambled peptides (Supplementary Fig. 2b), indicating that sialokinin binding to neurokinin receptors and subsequent immunomodulation is dependent on the protein sequence.

### Sialokinin activates the PI3K/Akt signaling pathway through NK₁R and NK₂R in human monocytes

We next performed correlation analysis to determine the relationship between the identified DEGs, and we observed that the expression levels of the seven downregulated DEGs were negatively correlated with *FAM120A* (Fig. 1c). *FAM120A* encodes an essential scaffolding protein that enables Src family kinases to phosphorylate and activate phosphatidylinositol 3-kinase (PI3K). Since NK₁R is known to activate PI3K[37], we hypothesized that sialokinin might alter the functions of monocytes through the PI3K signaling pathway. To investigate this, we examined PI3K and Akt activation in sialokinin-treated primary human

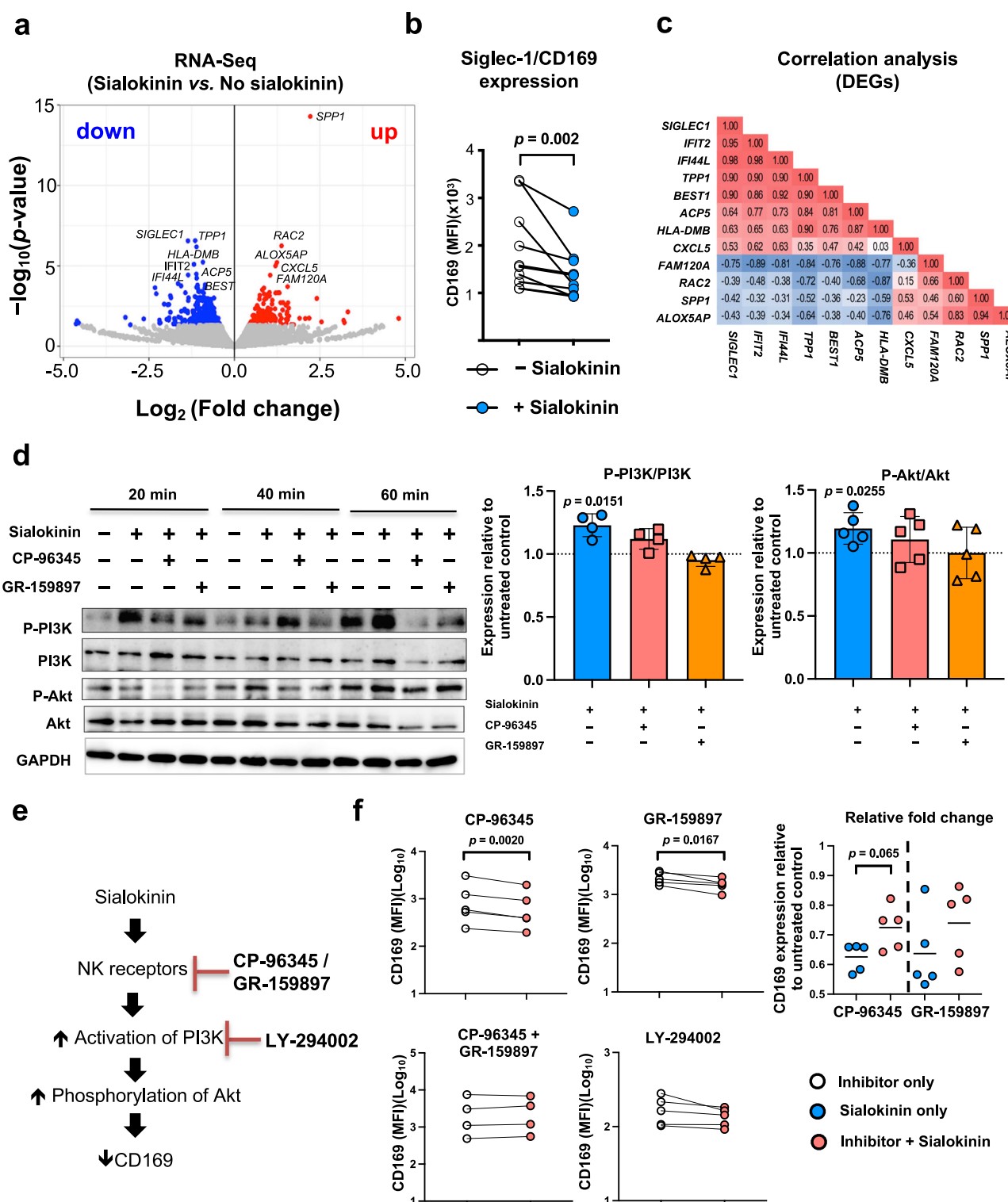

monocytes by western blot analysis. Sialokinin treatment stimulated phosphorylation of PI3K and Akt at 20 min post-treatment (Supplementary Fig. 3), and this effect was reduced by either an $NK_1R$ antagonist (CP-96345) or an $NK_2R$ antagonist (GR-159897) (Fig. 1d), indicating that sialokinin activates the PI3K/Akt signaling pathway through $NK_1R$ and $NK_2R$ in human monocytes.

We subsequently examined the downstream signaling linkage between the NK receptor/PI3K/Akt pathway and sialokinin-mediated regulation of monocyte activation by pre-treating cells with NK receptor antagonists (CP-96345 or GR-159897) or with the PI3K

inhibitor LY-294002 (Fig. 1e). Individually, the NK receptor antagonists only partially neutralized the sialokinin-mediated downregulation of Siglec-1/CD169 expression in human monocytes, while the PI3K inhibitor completely blocked this downregulation (Fig. 1f). To further investigate the contribution of $NK_1R$ and $NK_2R$, additional experiments were performed where monocytes were pre-treated with both CP-96345 and GR-159897 prior to sialokinin stimulation. As shown in Fig. 1f, dual receptor blockade resulted in a complete restoration of CD169 expression to baseline levels, comparable to PI3K inhibition. These findings suggest that PI3K signaling functions downstream of

**Fig. 1 | Sialokinin modulates monocyte activation via NK receptors and the PI3K/Akt signaling pathway. a** RNA-Seq profiling of primary human monocytes ($n = 3$) at 24 h post-treatment. Volcano plot indicates differentially expressed genes (DEGs) between sialokinin-treated vs. non-treated monocytes. DEG analysis was performed using edgeR, with multiple testing corrected via the Benjamini–Hochberg FDR method. Significantly upregulated and downregulated genes are shown in red and blue, respectively. **b** Flow cytometry shows reduced surface expression of Siglec-1/CD169 in sialokinin-treated monocytes compared with untreated monocytes ($n = 10$ biological replicates) at 24 h post-treatment. Data were analyzed by paired two-tailed *t*-test. **c** Pearson correlation matrix for the expressions of 12 significant DEGs from RNA-Seq data ($n = 3$). **d** Western blot analysis shows that sialokinin stimulation activated the PI3K-Akt pathway in human monocytes, with increased levels of phosphorylated PI3K (P-PI3K) and Akt (P-Akt) at 20, 40, and 60 min. Pre-treatment with either the NK$_1$R inhibitor CP-96345 or the NK$_2$R inhibitor GR-159897 led to inhibition of PI3K and Akt as early as 20 min post-stimulation. Relative expression levels of P-PI3K/PI3K ($n = 4$ biological replicates) and P-Akt/Akt ($n = 5$ biological replicates) in monocytes at the 20-min time point were quantified with ImageJ. Levels are expressed as fold change relative to the non-treated controls. Bar graphs represent the mean ± standard deviation (SD). Statistical comparisons between sialokinin-treated sample and untreated control were performed using paired two-tailed *t*-test. **e** Schematic representation of CD169 downregulation in response to sialokinin via NK receptor/PI3K/Akt activation. **f** The monocytes were pre-treated with the NK$_1$R antagonist (CP-96345; $n = 5$ biological replicates), NK$_2$R antagonist (GR-159897; $n = 5$ biological replicates), both antagonists in combination ($n = 4$ biological replicates), or the PI3K inhibitor (LY-294002; $n = 5$ biological replicates) prior to sialokinin stimulation. CD169 levels at 24 h post-treatment were measured by mean fluorescence intensity (MFI) and expressed as fold change relative to non-treated controls. Statistical comparisons were performed using paired two-tailed *t*-test or Wilcoxon signed-rank test, depending on data distribution.

NK$_1$R and NK$_2$R activation, and that both receptors are required for sialokinin-mediated modulation of monocyte activation.

### Sialokinin limits myeloid cell activation following CHIKV infection

Monocytes are the primary immune cells that react to CHIKV infection in humans[38], and our ex vivo virus infection model using isolated human PBMCs also showed that monocytes were the predominant immune cell subset that were infected and exhibited elevated CD169 expression after CHIKV infection (Supplementary Fig. 4). Therefore, we next assessed whether sialokinin would impact the levels of activated monocytes or macrophages following CHIKV infection.

To determine the optimal time points for assessing CHIKV infectivity and CD169 expression following sialokinin treatment, kinetic experiments were performed at 24- and 48-h post-infection (hpi) using both human monocytes and monocyte-derived macrophages (MDMs). These experiments revealed that 24 hpi is optimal for monocytes, while 48 hpi is optimal for MDMs, based on the observed differences in magnitude of infection and CD169 expression levels with sialokinin treatment (Supplementary Figs. 5 and 6). These time points were subsequently used in all downstream analyses.

We first focused on human monocytes to assess whether sialokinin modulates their activation and susceptibility to CHIKV infection in cells isolated from additional healthy donors. Human monocytes were treated with sialokinin (10 µM) and infected with CHIKV at a multiplicity of infection (MOI) of 10 for 24 h. We observed significant activation of monocytes following CHIKV infection, as reflected by upregulation of CD169 surface expression, and this activation was significantly reduced by sialokinin treatment (Fig. 2a). In parallel, CHIKV infectivity was also significantly decreased in the presence of sialokinin and exhibited lower viral loads (Fig. 2b). Total RNA was extracted from harvested monocytes and analyzed by quantitative real-time PCR. Expression levels of key interferon-stimulated genes (ISGs), including *Siglec1*, *IFIT2*, and *IFI44L*, were measured and normalized to housekeeping gene expression using the ΔΔCt method. Notably, sialokinin treatment significantly suppressed the expression of all three ISGs not only in non-infected monocytes but also under CHIKV-infected conditions (Supplementary Fig. 7).

To further investigate the role of sialokinin, we conducted pre-treatment experiments in which human monocytes were exposed to sialokinin prior to CHIKV infection. Notably, pre-treatment with sialokinin reduced both CHIKV infection and CD169 expression (Supplementary Fig. 8), further supporting the role of sialokinin in modulating monocyte susceptibility to CHIKV.

We then extended these observations to MDMs. At 48 hpi, we observed similar trends. CHIKV infection induced CD169 expression and viral replication in MDMs, both of which were attenuated by sialokinin treatment, although the magnitude of the effect was less prominent than that observed in monocytes (Fig. 2a, b). These findings suggest that sialokinin consistently dampens CHIKV-induced activation and infection across monocyte-lineage cells.

### Sialokinin attenuates CHIKV pathogenesis in vivo

To determine whether reduced monocyte activation affects disease outcomes in vivo, we assessed the effect of sialokinin in adult mice infected with CHIKV[39]. In the sialokinin-treated group, mice were infected with CHIKV and received 1 µg of sialokinin via subcutaneous injection into the joint footpad, a dose corresponding to the amount found in mosquito salivary glands. The presence of sialokinin during CHIKV infection restricted monocyte activation in vivo, with treated mice showing a lower percentage of CD169+ monocytes in blood circulation following virus infection (Fig. 3a). These results align with our ex vivo data from primary human monocytes and macrophages. Given that sialokinin has been reported to enhance vascular leakage, we also measured circulating neutrophil levels to assess potential changes in vascular permeability and immune cell trafficking. At 3 days post-infection (dpi), we observed a trend toward higher circulating neutrophil counts in sialokinin-treated mice compared to untreated controls. However, this difference did not reach statistical significance (Supplementary Fig. 9a).

Compared to untreated CHIKV-infected mice, we observed a significant reduction in joint footpad swelling in sialokinin-treated mice at the peak of inflammation (6 dpi) (Fig. 3b). The sialokinin-treated group also demonstrated a lower level of viremia at 2 to 7 dpi compared to the untreated control group (Fig. 3c). No significant change was observed in either joint inflammation or viremia when the CHIKV-infected mice were given 1 µg of scrambled peptides (Supplementary Fig. 9b). A lower dose of 0.5 µg sialokinin also significantly decreased joint inflammation and viremia during CHIKV infection (Supplementary Fig. 9c).

To investigate whether sialokinin promotes early viral dissemination beyond the injection site, we measured viral titers in peripheral tissues at 1 dpi, including the contralateral (left) footpad, muscle, and spleen. Notably, we observed a significantly higher viral load in the contralateral footpad of sialokinin-treated mice compared to controls (Fig. 3d), suggesting that sialokinin facilitates early viral spread to distal tissues. However, no significant differences in viral titers were detected in the muscle or spleen at this early time point (Fig. 3d). These findings support the role of sialokinin in enhancing early viral dissemination, particularly to anatomically adjacent tissues.

### Sialokinin affects IFN-γ production by joint-infiltrating CD4+ T cells in

**CHIKV-infected mice.** We hypothesized that differences in joint inflammation may be attributable to differences in the profile of leukocytes infiltrating the joints. To investigate this, we first measured viral titers in the joint footpad tissue at 6 dpi and found no significant difference in viral load between the CHIKV-only and CHIKV + sialokinin

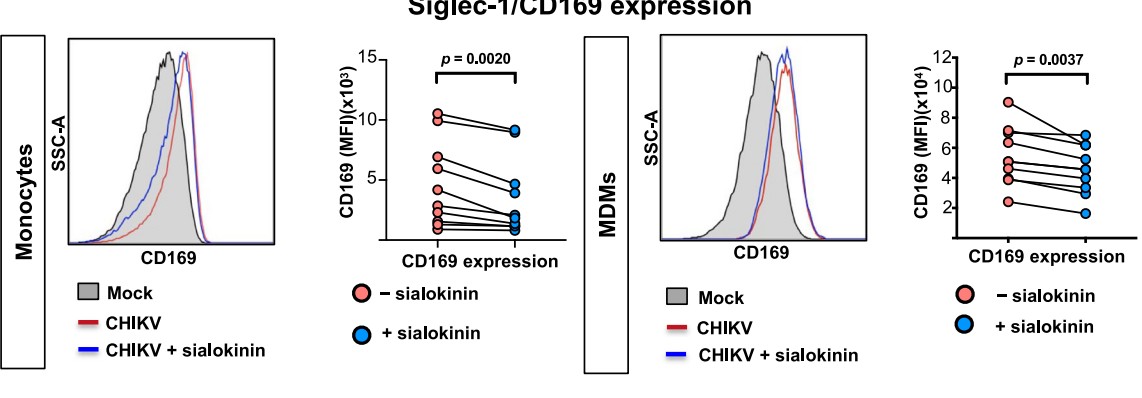

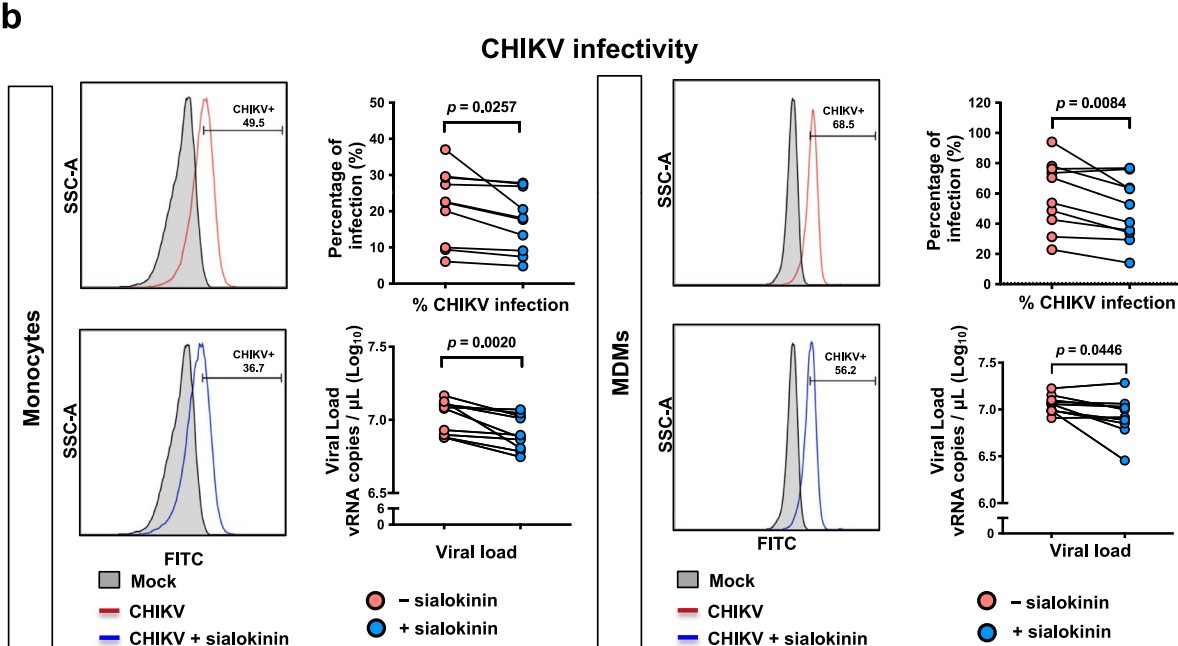

**Fig. 2 | Sialokinin reduces activation and CHIKV infection in primary human myeloid cells.** Human monocytes and monocyte-derived macrophages (MDMs) from ten healthy donors were infected with CHIKV in the presence or absence of sialokinin (10 μM). **a** Activation profiles of monocytes and MDMs were assessed by flow cytometry, focusing on CD169 expression. Sialokinin treatment significantly reduced CD169 expression was lower in both monocytes and MDMs following CHIKV infection (MOI = 10) at 24 (monocytes) or 48 (MDMs) hours post-infection.

**b** CHIKV infection was determined using FACS analysis and qRT-PCR on the viral supernatant. Viral load was quantified by qRT-PCR targeting CHIKV nsP1 viral copies. Data were $log_{10}$-transformed prior to statistical analysis. CHIKV infectivity toward monocytes and MDMs was reduced in the presence of sialokinin. Statistical analyses were performed using either a paired two-tailed $t$-test or a Wilcoxon signed-rank test, depending on data distribution.

groups (Fig. 3e), suggesting that the observed immune modulation is not due to differences in local viral replication. We then harvested joint footpad cells at 6 dpi and characterized immune cell profiles by flow cytometry. After quantifying the cell counts of major leukocyte subsets, including neutrophils, natural killer cells, macrophages, CD4+ and CD8+ T cells, we observed no significant differences between the two groups (Supplementary Fig. 9d). Since CD4+ T cells mediate CHIKV-induced footpad swelling at the peak of joint inflammation[39,40], variations in the infiltrating T cells were examined between groups. Their capacity to release several cytokines (including IFN-γ, TNF-α, IL-10, IL-4, and IL-17) in response to non-specific stimulation by PMA and ionomycin were assessed. In mice receiving sialokinin, fewer IFN-γ-producing CD4+ T cells were detected upon activation (Fig. 3f).

To directly assess virus-specific responses, we performed ELISpot assays on CD4+ T cells isolated from the joint footpad at 6 dpi. These cells were stimulated ex vivo with CHIKV virions in the presence of IL-2 to promote antigen-specific cytokine production. Consistent with the intracellular cytokine staining results, we observed a significant reduction in the frequency of CHIKV-specific IFN-γ-producing CD4+ T cells in sialokinin-treated mice (Fig. 3g).

**Treatment with sialokinin reduces CHIKV-induced joint inflammation**

Given the anti-inflammatory phenotype elicited by sialokinin in vivo, we hypothesized that this peptide could be used to treat CHIKV-induced joint inflammation. To evaluate its therapeutic effect after a viral infection, CHIKV-infected mice were treated with sialokinin (1 mg/kg) daily starting at 2 dpi and the development of joint pathology was assessed (Fig. 4a). As sialokinin has been reported to enhance vascular leakage, we were concerned that systemic administration via

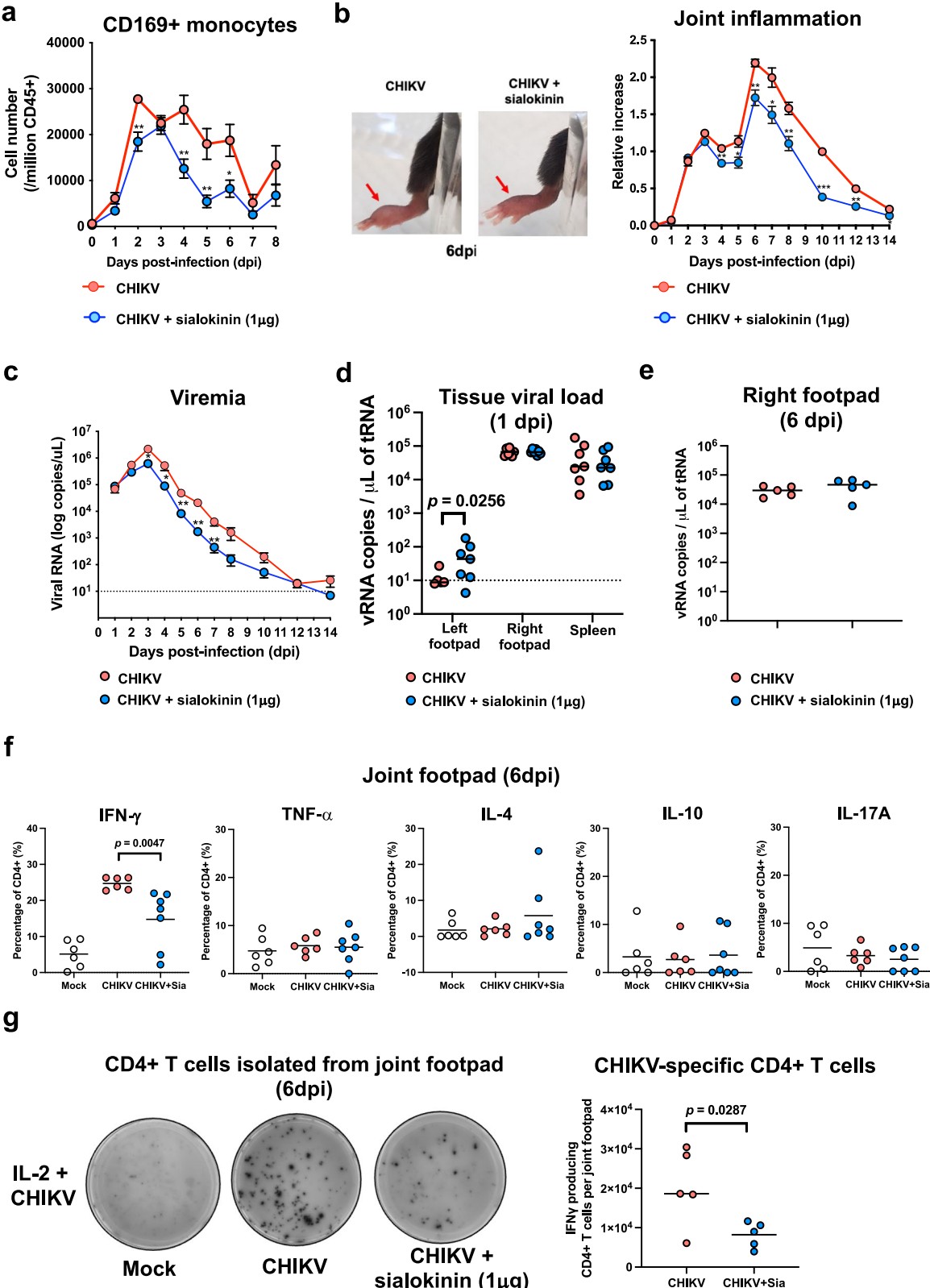

intraperitoneal (IP) injection might exacerbate this effect. To address this, we measured circulating neutrophil levels as an indicator of potential changes in vascular permeability and immune cell trafficking. However, no significant difference in neutrophil counts was observed between sialokinin-treated and untreated mice (Fig. 4b), suggesting that sialokinin administration does not induce significant systemic vascular leakage. Following treatment, sialokinin-treated mice showed

a lower number of CD169⁺ monocytes in blood circulation at 4 and 6 dpi (Fig. 4b), and significantly reduced joint footpad swelling at the peak of infection (7 dpi), though the treated mice did not show a significant difference in viremia compared to untreated controls (Fig. 4c).

To further investigate the local immune response, we measured viral titers in the joint footpad tissue at 6 dpi and found no significant difference in viral load between the treated and untreated groups

**Fig. 3 | Sialokinin reduces circulating CD169⁺ monocytes and joint inflammation in CHIKV-infected animals.** C57BL/6 WT 4-week-old (CHIKV, $n = 8$; CHIKV + sialokinin, $n = 10$) mice were infected with $1 \times 10^6$ pfu of CHIKV subcutaneously at the joint footpad. Sialokinin (1 µg) was administered concurrently with CHIKV in the treatment group. **a** CD11b⁺ Ly6C⁺ CD169⁺ monocytes in blood of CHIKV-infected mice were monitored daily by flow cytometry. **b** Representative images at 6 dpi show reduced joint footpad swelling in sialokinin-treated mice. Arrows indicate the area of persistent swelling. Joint swelling was monitored over 14 dpi. **c** Viremia was assessed from tail vein blood (1 to 10 dpi) via qRT-PCR targeting CHIKV nsP1. Data were collated from two independent experiments and are shown as mean ± SEM, analyzed using two-tailed $t$-test. **d** At 1 dpi, tissue viral loads were quantified by qRT-PCR of CHIKV nsP1 viral copies in the contralateral left joint footpad, the right joint footpad and the spleen of CHIKV-infected mice (CHIKV, $n = 7$; CHIKV + sialokinin,

$n = 7$). Statistical comparisons used unpaired two-tailed $t$-test or Mann–Whitney $U$ test, depending on data distribution. **e** At 6 dpi, viral loads in the infected right joint footpad were measured by qRT-PCR of CHIKV nsP1 viral copies (CHIKV, $n = 5$; CHIKV + sialokinin, $n = 5$) and analyzed using unpaired two-tailed $t$-test. **f** CD4⁺ T cell cytokine profiles (IFN-γ, TNF-α, IL-4, IL-10, and IL-17A) were assessed at 6 dpi in joint footpads of CHIKV-infected (CHIKV, $n = 6$; CHIKV + sialokinin, $n = 7$) and mock ($n = 6$) mice after PMA/ionomycin stimulation. Data are presented as mean ± SD, and analyzed using one-way ANOVA with Tukey's post-hoc test. **g** Representative images of ELISpot wells depicting the number of IFN-γ-producing cells in enriched CD4⁺ T cells from joint footpads at 6 dpi. Quantitative plots show the total numbers of IFN-γ-producing CD4+ T cells per infected joint footpad. Statistical comparison between the two groups was performed using unpaired two-tailed $t$-test.

(Fig. 4c), indicating that the reduced inflammation was not due to altered viral replication at the site of pathology. Except for a reduced quantity of neutrophils in mice receiving sialokinin therapy, we found no significant variations in major immune cells in joint footpad cells harvested at 6 dpi (Fig. 4d). However, the sialokinin-treated group had significantly lower percentages of IFN-γ-producing T cells in the joint footpad (Fig. 4e). To directly assess virus-specific T cell responses in the therapeutic setting, we performed ELISpot analysis on CD4⁺ T cells from the joint footpad at 6 dpi. Consistent with earlier findings, sialokinin-treated mice exhibited a significant reduction in CHIKV-specific IFN-γ-producing CD4⁺ T cells (Fig. 4e), reinforcing its role in dampening local antiviral T cell responses.

This reduction in T cell activation was accompanied by lower levels of pro-inflammatory cytokines, including IL-2, IL-27, IL-15, and IL-28, in the joint footpad tissue of sialokinin-treated mice (Fig. 4f), further supporting the anti-inflammatory effect of sialokinin during CHIKV infection.

### Human IgG levels against sialokinin correlated with disease severity and inflammation in CHIKV patients

Mosquito saliva is known to trigger an antibody response in humans following mosquito bites[41], and anti-salivary protein antibody levels have been associated with the risk of mosquito-borne diseases[42]. Given the immunomodulatory and anti-inflammatory properties observed in our characterization, we investigated the potential antibody response to sialokinin and its association with disease progression in human CHIKV by assessing the levels of anti-sialokinin antibodies in patients infected with CHIKV[43]. Analysis of antibody responses to the sialokinin peptide in plasma samples collected during the acute phase (median 4 days post-illness onset) from 30 CHIKV patients revealed a wide range of sialokinin-specific IgG levels, varying from 2 to more than 30 times the level observed in pooled healthy donor samples. Notably, ~97% (29 out of 30) of the patients demonstrated IgG positivity against sialokinin (Fig. 5a). A higher antibody response was observed in patients with severe disease compared to those with mild illness ($p = 0.063$) and a significant positive correlation between viral load and IgG levels against sialokinin (rho = 0.500, $p = 0.006$) (Fig. 5b). Additionally, positive correlations were observed between systemic inflammatory markers (CRP, IL-1RA, IL-6, IFN-α, IL-12, MCP-1, IL-15, IP-10, and MIP-1β) and IgG antibodies against sialokinin (Fig. 5b, c).

### Discussion

While mosquito saliva has demonstrated significant effects on arthropod-borne virus (arbovirus) infection, our understanding of the molecular mechanisms governing facilitation of virus transmission and disease pathogenesis remains limited, primarily due to sparse functional research on mosquito salivary proteins. Accumulated evidence indicates that mosquito salivary proteins exert pleiotropic effects by directly interacting with pathogens or indirectly modulating the host immune response to enhance virus infections. Notably, several *Aedes* salivary proteins (AAEL000793, AAEL007420, and AAEL006347) have

been shown to bind to the flavivirus Zika virus envelope protein, although their role in virus transmission remains unclear[44]. Other salivary gland proteins, such as apyrase, neutrophil-stimulating protein 1 (NeSt1), and D7 can interact with human immune cell receptor proteins, which could impact mosquito-borne viral infections like dengue, Zika, and chikungunya[45]. *A. aegypti* venom allergen-1 (AaVA-1) specifically promotes dengue and Zika virus transmission by activating autophagy in host immune cells of the monocyte lineage[46]. These findings collectively indicate that mosquito proteins have significant pharmacological functions.

The sequence similarities between mammalian tachykinins and sialokinin, originally identified as a vasodilator in mosquitoes, prompted us to investigate its potential immunomodulatory properties. In mice, sialokinin enhances the spread of viruses by causing vascular leakage[32] and orchestrates a shift in mammalian immunity towards an anti-inflammatory CD4⁺ T helper 2 (Th2) response[47]. In our study, we revealed the immunomodulatory effect of sialokinin on the myeloid cell population, which plays a pivotal role in the pathogenesis of CHIKV infections.

Early in infection, we observed increased viral load in the contralateral (left) footpad at 1 dpi in sialokinin-treated mice, indicating that sialokinin facilitates systemic virus dissemination. This supports the hypothesis that mosquito-derived sialokinin promotes virus spread beyond the inoculation site via vascular leakage. Interestingly, despite this early enhancement of virus dissemination, viral titers in the inoculated footpad were comparable between groups by 6 dpi. However, we detected a significant reduction in virus-specific CD4⁺ T cell responses in sialokinin-treated mice. This apparent discrepancy likely reflects the temporal dynamics between virus replication and immune activation. While virus replication may have plateaued or declined by 6 dpi, early sialokinin-driven modulation of monocytes had already shaped the downstream T cell response.

We propose that sialokinin exerts its immunomodulatory effects by suppressing monocyte activation during the early phase of infection. Monocytes are key antigen-presenting cells, and their activation status directly influences cytokine production and T cell priming. In our study, sialokinin treatment led to a marked reduction in monocyte and macrophage activation, as evidenced by decreased Siglec-1/CD169 expression. Importantly, we also observed a reduction in circulating activated monocytes, suggesting that sialokinin's immunosuppressive effects extend beyond the local tissue environment to influence systemic immune responses. This early suppression likely impairs antigen presentation and cytokine signaling, resulting in diminished virus-specific CD4⁺ T cell priming and differentiation.

Activated monocytes with enhanced CD169 expression were previously reported in the peripheral circulation of rhesus macaques following CHIKV infection and are associated with CHIKV-induced joint inflammation[48]. Sialokinin significantly restricted the activation of monocytes and macrophages, as indicated by decreased Siglec-1/CD169 expression. Unlike M1 and M2 macrophages, CD169⁺ macrophages interact directly with T cells, B cells, and dendritic cells to

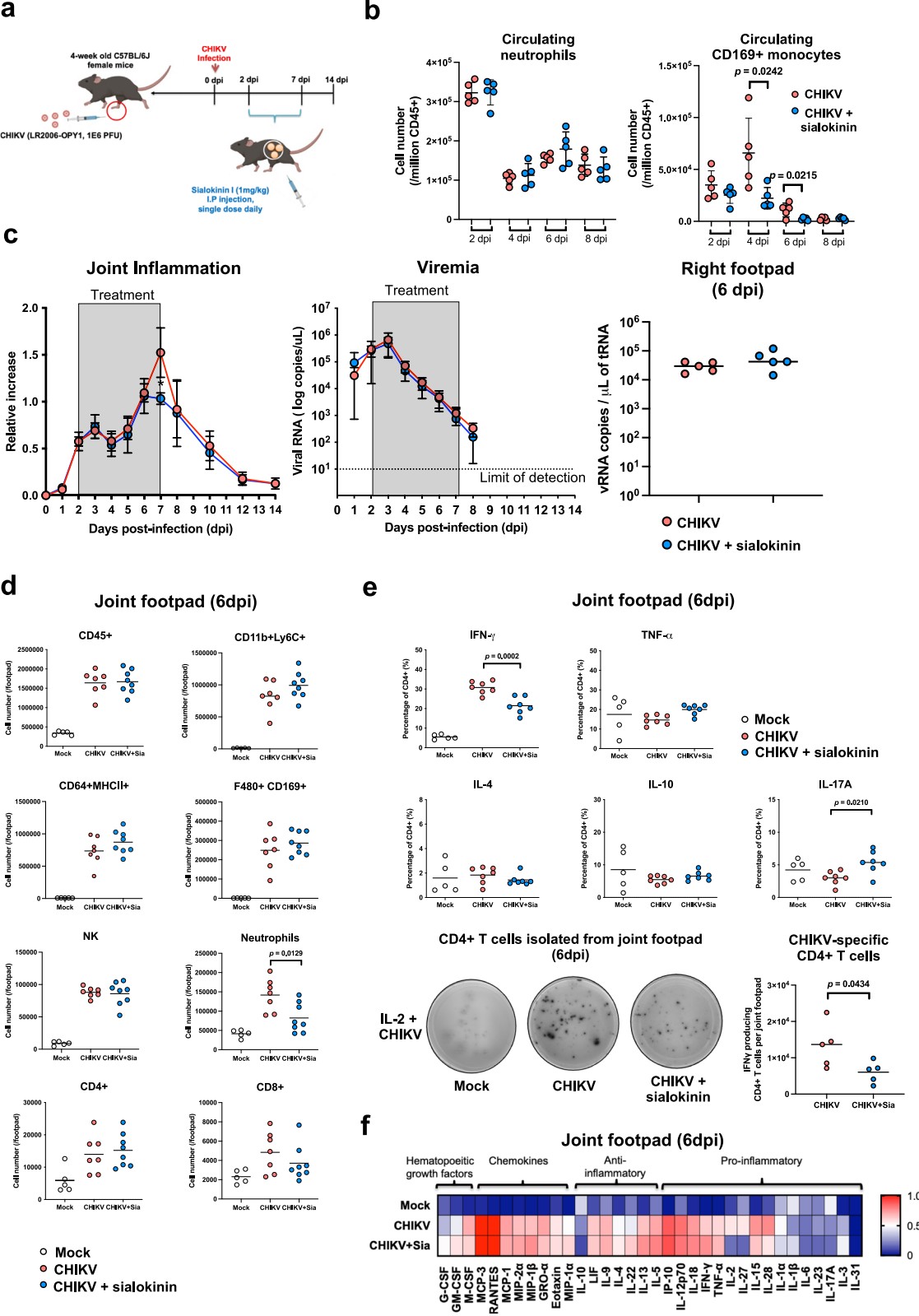

participate in immune regulation[49]. Delivery of antigen to CD169+ macrophages has been implicated in enhancing CD4+ T cell priming in mice[50]. In our study, sialokinin treatment led to a significant reduction in CD169+ monocytes and a profound reduction in IFN-γ producing CD4+ T cell response in the joint footpad of CHIKV-infected mice, highlighting the potential association between CD169+ macrophages and the activation of adaptive T cell responses during CHIKV infection.

Our findings complement a prior study showing immune crosstalk between infiltrating CD4+ T cells and myeloid cells in CHIKV-infected joints, with depletion of the CD64+ macrophages significantly reducing joint inflammation during CHIKV infection[51].

The adaptive immune response to CHIKV infection may be a double-edged sword—while it is essential for the clearance of infectious viruses, it also contributes to disease pathology during the acute

**Fig. 4 | Sialokinin treatment reduces joint footpad inflammation in CHIKV-infected mice. a** C57BL/6 WT 4-week-old mice were infected with $1 \times 10^6$ pfu of CHIKV subcutaneously at the joint footpad. Mice were treated with 1 mg/kg of sialokinin or PBS (control) via intraperitoneal injection from days 2 to 7 post-infection (dpi). Created in BioRender. Ng, L. (2025) https://BioRender.com/fj6toz3. **b** Circulating CD11b$^+$Ly6G$^+$ neutrophils and CD11b$^+$ Ly6C$^+$CD169$^+$ monocytes were quantified by flow cytometry. Data are presented as mean ± SD ($n = 5$ per group). Statistical comparisons used unpaired two-tailed $t$-test. **c** Joint swelling was monitored to 14 dpi. Viremia was assessed from tail vein blood (1 to 10 dpi) via qRT-PCR targeting CHIKV nsP1. Data are presented as mean ± SD ($n = 5$ per group). *$p < 0.05$, **$p < 0.01$, ***$p < 0.001$ by unpaired two-tailed $t$-test. Tissue viral loads were measured at 6 dpi in the right joint footpad (CHIKV, $n = 5$; CHIKV + sialokinin, $n = 5$). Statistical comparisons were performed using unpaired two-tailed $t$-test. **d** Immune cell subsets in the joint footpad were analyzed at 6 dpi (PBS, $n = 7$; sialokinin, $n = 7$; mock, $n = 5$). Data were analyzed by one-way ANOVA with Tukey's post-hoc test. **e** CD4$^+$ T cell cytokine profiles (IFN-γ, TNF-α, IL-4, IL-10, and IL-17A) were assessed at 6 dpi in the joint footpads of CHIKV-infected (PBS, $n = 7$; sialokinin, $n = 7$) and mock ($n = 5$) mice after PMA/ionomycin stimulation. Data are presented as mean ± SD and analyzed using one-way ANOVA with Tukey's post-hoc test. CD4+ T cells isolated from right joint footpads at 6 dpi ($n = 5$ per group) were stimulated with CHIKV for ELISpot assay. Representative images and quantitative plots show reduced CHIKV-specific IFN-γ-producing CD4+ T cells in the joint footpads of treated mice at 6 dpi. Statistical comparison used unpaired two-tailed $t$-test. **f** Cell lysates from the joints of CHIKV-infected (PBS, $n = 7$; sialokinin, $n = 7$) and mock ($n = 5$) mice at 6 dpi were analyzed using a 36-plex microbead-based immunoassay. Immune mediators are grouped based on function. Heatmap colors indicate relative concentration (blue = low, red = high).

phase. The absence of an adaptive immune response in CHIKV-infected mice resulted in lower peak joint footpad swelling, implicating T and/or B cells in the early stages of immune pathogenesis[40,52]. In our in vivo experiments, treating CHIKV-infected mice with sialokinin reduced monocyte activation, accompanied by decreased swelling and pathogenic CD4$^+$ Th1 cells at the footpad during the peak of inflammation. These findings suggest that sialokinin could be used as an alternative immunomodulatory therapy to abrogate CHIKV-induced joint inflammation.

Beyond CHIKV, activated myeloid cells, particularly CD169$^+$ macrophages, play important roles in human inflammatory diseases, including multiple sclerosis[53], rheumatoid arthritis[54], and chronic obstructive pulmonary disease (COPD)[55]. In this study, we identified key human receptors and signaling pathways that govern the activation of human monocytes and macrophages in response to sialokinin. Our in vitro data demonstrate that both NK$_1$R and NK$_2$R serve as primary receptors mediating the immunomodulatory effects of sialokinin, acting through PI3K/Akt signaling to suppress monocyte activation. This discovery provides an avenue in targeting the NK receptor or its downstream signaling pathway to ameliorate CD169$^+$ macrophage-mediated inflammation. Future studies employing double knockout mouse models for NK$_1$R and NK$_2$R will be valuable for exploring the therapeutic potential of sialokinin and its derivatives in treating chronic inflammatory diseases, including chikungunya.

An earlier study employing sialokinin knockout mosquitoes reported decreased levels of total leukocytes, neutrophils, and CD8$^+$ T cells in the footpads of BALB/c mice following mosquito bites[56]. Subsequent experiments using sialokinin knockout saliva in humanized mice demonstrated the induction of a Th1 cellular immune response, indicating that the sialokinin peptide triggers a Th2 cellular immune response during wild-type mosquito biting[47]. However, when we injected sialokinin into C57BL/6 mice as a therapeutic after CHIKV infection, we observed a reduced Th1 response, but no induction of a Th2 response. This phenomenon likely relates to the inherent predisposition of C57BL/6 mice toward a Th1 immune response[57,58]. The downregulation of CD169 in monocytes exposed to sialokinin provides evidence that mosquito salivary sialokinin promotes the polarization of monocytes toward an immunoregulatory, deactivated phenotype. This phenotype is likely advantageous for arboviruses as it interferes with the activation of other immune cells, particularly the CD4$^+$ T cells responsible for virus clearance.

Taken together, our findings underscore the multifaceted role of sialokinin in CHIKV pathogenesis. It facilitates early virus dissemination, suppresses monocyte activation both locally and systemically, and impairs virus-specific CD4$^+$ T cell responses. These insights expand upon previous work and highlight sialokinin's potential as a therapeutic target for modulating inflammation in CHIKV and other immune-mediated diseases.

Sialokinin is immunogenic in CHIKV-infected patients, with IgG levels correlating with disease severity and inflammation. Our findings suggest that sialokinin plays an anti-inflammatory role; however, individuals frequently exposed to mosquitoes could potentially develop higher levels of anti-sialokinin antibodies, which may interfere with sialokinin function and contribute to increased inflammation, potentially worsening the severity of CHIKV infection. Understanding the complex interactions between the virus, vector, and host is crucial for refining pathogenesis models, improving strategies to reduce disease transmission, and advancing therapeutics and transmission-blocking vaccines. This is particularly important in endemic regions, where ongoing exposure to both infected and uninfected mosquitoes complicates disease control efforts. Notably, our human patient cohort had a sex imbalance (26 males vs. 4 females), which may influence immune responses and disease severity. While our key findings remain robust, future studies with more balanced cohorts are needed to assess potential sex-specific differences in antibody responses to mosquito saliva components and their impact on disease outcomes.

As climate change continues to reshape ecosystems, the prevalence and geographical range of mosquito-borne infections are expected to expand, further intensifying the global health burden. Our study demonstrated the importance of understanding immune regulation induced by a particular mosquito salivary ligand in CHIKV immunopathology. It also highlights new avenues for identifying potential targets to combat mosquito-borne viral infections and associated inflammatory diseases. Targeting these factors presents a promising strategy to complement traditional approaches, offering new ways to combat the rising threat of mosquito-borne diseases in a warmer world.

## Methods
### Human blood and ethics approval
Human blood apheresis cones were obtained from 35 healthy adult donors with written informed consent, in accordance with guidelines from the Health Sciences Authority of Singapore (IRB No: 2017/2512). Sampling was approved specifically for the collection of samples from healthy donors. For the CHIKV cohort in Singapore, written informed consent was obtained from all participants. The study was approved by the National Healthcare Group's domain-specific ethics review board (DSRB No. B/08/026).

### Patients and plasma collection
The CHIKV cohort study included 30 chikungunya fever patients admitted to the Communicable Disease Centre at Tan Tock Seng Hospital from 1 August through 23 September 2008. The outbreak occurred in an industrial area where the source population comprised predominantly male workers, with only 4 female individuals among the 30 patients in this study. Participants ranged in age from 23 to 67 years, with a median age of 36.5 years. Plasma specimens were collected on the day of admission to the hospital (acute phase; median, 4 days after illness onset). All patients were confirmed to have CHIK fever by reverse-transcription polymerase chain reaction (RT-PCR). Illness was

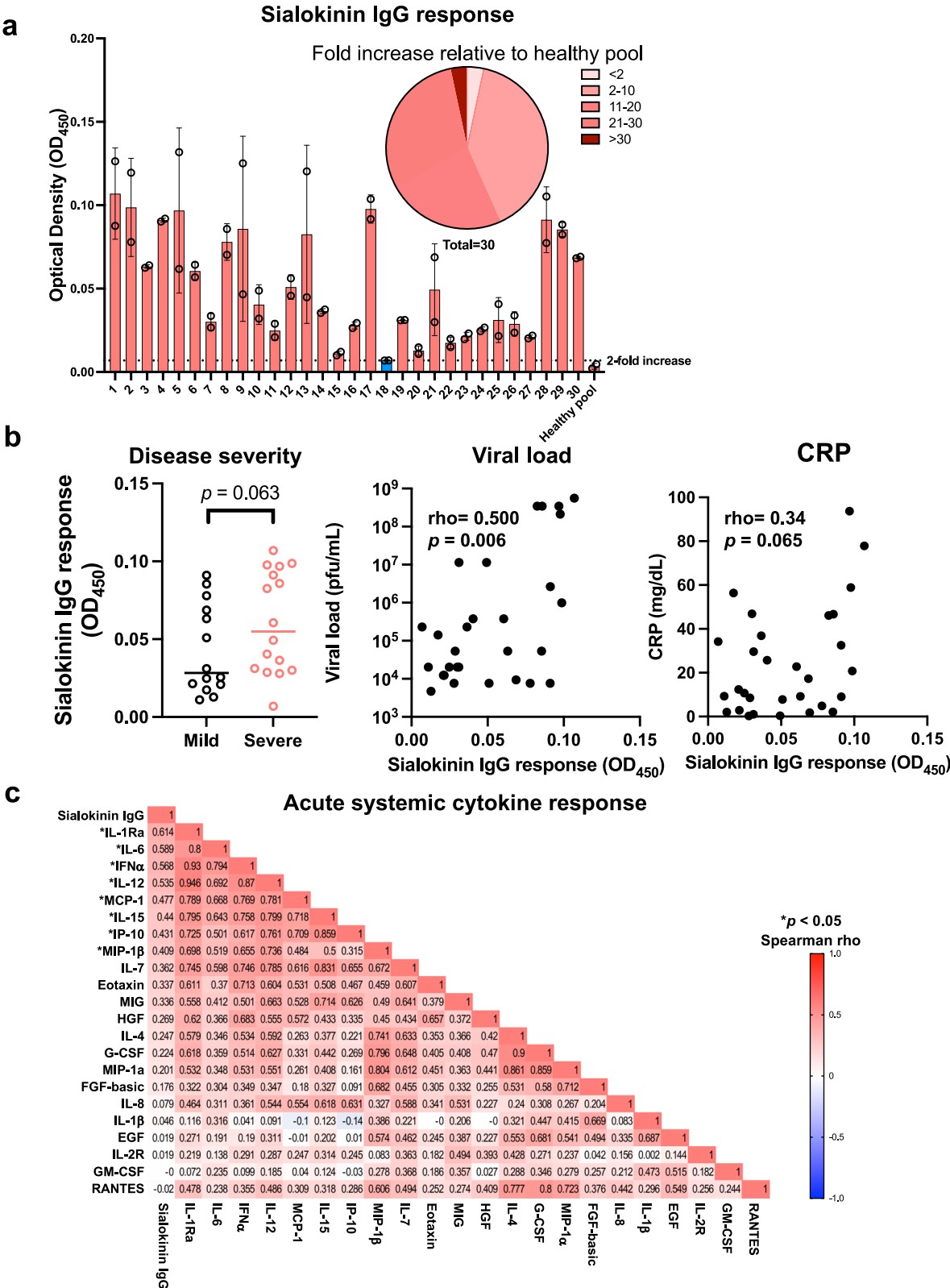

**Fig. 5 | Sialokinin-specific IgG response in CHIKV patient cohort. a** IgG levels against sialokinin were measured via ELISA in acute samples (median 4 days after illness onset) from 30 patients with confirmed CHIKV infection. Data are presented as mean ± SD. The pie chart illustrates the proportion of patients exhibiting different fold increases in sialokinin IgG compared to the healthy pooled sample. **b** The associations between sialokinin IgG levels and disease severity, viral load, and systemic inflammatory CRP levels in acute CHIKV patient samples. Statistical analyses were conducted using the two-tailed Mann–Whitney $U$ test and Spearman rank correlation analysis. **c** The correlation between sialokinin IgG levels and the acute systemic cytokine response in CHIKV patients. Immune mediator concentrations were quantified using microbead-based immunoassay. The heatmap displays Spearman's rank correlation coefficient values for each immune mediator and the sialokinin IgG response. Statistical analyses were performed using two-tailed Spearman rank correlation analysis, with significance indicated by *$p < 0.05$.

defined as severe if a patient had either a maximum temperature >38.5 °C, a maximum pulse rate >100 beats/min, or a nadir platelet count <$100 \times 10^9$ cells/L. Patients who do not fulfil these criteria were classified as having mild illness. The clinical data, viral load data, and systemic cytokine data used for association and correlation analysis were obtained from published datasets[43] deposited by the cohort.

### Peptides and antagonists
*Aedes aegypti* sialokinin I peptides were synthesized by solid-phase synthesis using a CEM Liberty Blue automated microwave peptide synthesizer. The purity of the peptides was checked by HPLC and was > 95%. The peptides were dissolved in phosphate-buffered saline (PBS) and stored at a concentration of 5 mM at −20 °C. CP-96345 (NK$_1$R antagonist) was purchased from Sigma-Aldrich (Darmstadt, Germany) and GR-159897 (NK$_2$R antagonist) was from Santa Cruz Biotechnology (Texas, USA). The antagonists were prepared in DMSO and stored as 10 mM (CP-96345) and 50 mM (GR-159897) stock solutions.

### Cells and virus stocks
Chinese hamster ovary (CHO) cells expressing three distinct types of human NK receptors (NK$_1$R, NK$_2$R, and NK$_3$R) were a generous gift from Dr Shinya Oishi, Faculty of Pharmaceutical Sciences, Kyoto University, Japan. These stably transfected cells were cultured in 10% FBS/ Ham's F12 media supplemented with hygromycin (500 µg/mL) and penicillin-streptomycin (100 U/mL). Cultures were maintained at 37 °C in a humidified atmosphere of 5% $CO_2$.

Human peripheral blood mononuclear cells (PBMCs) were isolated from human whole blood using Ficoll gradient centrifugation. Monocytes were purified through negative selection using an indirect magnetic labeling system (Monocyte Isolation Kit II; Miltenyi Biotec). Isolated monocytes were incubated in complete Iscove's modified Dulbecco's medium (IMDM) (HyClone) supplemented with 10% (vol/ vol) heat-inactivated human serum (HS) (Sigma-Aldrich) and 1% (vol/ vol) penicillin-streptomycin for 6 days to differentiate into monocyte-derived macrophages (MDMs). Medium was changed on day 3 of differentiation.

LR2006 OPY1 (the CHIKV strain originally isolated from a French patient returning from Réunion Island during the 2006 CHIKF outbreak)[59] and ZsGreen-tagged LR2006 OPY1 were used in this study. Virus stocks used for ex vivo experiments were produced in Vero-E6 cells. For in vivo experiments, viruses were propagated in C6/36 cells. Titers were determined by standard plaque assays using Vero-E6 cells.

### Calcium flux assay
Calcium flux assay was carried out on CHO cells stably transfected with one of the human NK receptors (NK$_1$R, NK$_2$R, or NK$_3$R). CHO cells were seeded at a density of 70,000 cells/well in 100 µL of Ham's F-12 media in a black 96-well clear-bottom plate (Greiner). After overnight incubation, the media was replaced with Calcium 6 assay dye from Molecular Devices (1:1 dilution with buffer solution of 1× HBSS containing 20 mM HEPES) followed by incubation for 2 h at 37 °C for the absorption of dye into the cytoplasm. The intracellular calcium ion flux analysis was performed at different concentrations of sialokinin (ten-fold dilution) using a FLIPR Tetra instrument (Molecular Devices). Fluorescence signal was recorded at excitation and emission wavelengths of 485 nm and 535 nm, respectively. The change in fluorescence is presented as the percent change with respect to baseline fluorescence. EC$_{50}$ values were determined from dose-response curves plotted using Prism 5 (GraphPad).

For the dose-response study in the presence of a competitive inhibitor, competitive inhibition was performed using NK$_1$R antagonist CP-96345 and NK$_2$R antagonist GR-159897. Following incubation of the cells for 30 min with different concentrations of the antagonist (50, 500, or 5000 nM), the calcium flux assay was performed using sialokinin I [at 8 different concentrations from 0.1 pM to 1 µM (NK$_1$R and

NK$_2$R) or 1 pM to 10 µM (NK$_3$R) depending on the receptor type)]. Data were obtained from the average of three repetitions performed in triplicate for each concentration.

### Sialokinin, NK$_1$R/NK$_2$R antagonist and PI3K inhibitor treatment
Freshly purified human monocytes and six-day-differentiated MDMs ($1 \times 0^6$ cells/well) were incubated with or without sialokinin peptides in complete IMDM supplemented with 10% (vol/vol) HS (Sigma-Aldrich) and 1% (vol/vol) penicillin-streptomycin. Untreated monocytes and MDMs served as no-treatment controls. NK$_1$R antagonist CP-96345, NK$_2$R antagonist GR-159897 and PI3K inhibitor LY-294002 were used to confirm the specific effects of sialokinin. The monocytes and MDMs were pre-treated with CP-96345 or GR-159897 or PI3K inhibitor LY-294002 for 30 min before the addition of sialokinin.

### Ex vivo virus infection
CHIKV infections in human whole blood were performed at a multiplicity of infection (MOI) of 10 with CHIKV strain LR2006 OPY1. The whole blood was incubated at 37 °C until harvesting at 24 h post-infection (hpi). The cells were stained for the detection of surface markers via flow cytometry. For CHIKV infection in primary monocytes and MDMs, cells were treated with sialokinin and infected with CHIKV strain LR2006 OPY1 tagged with ZsGreen at MOI = 10. In parallel, cells were also infected with CHIKV in the absence of sialokinin, and these infected but untreated cells served as controls. Cells were then harvested at 24 hpi (monocytes) or 48 hpi (MDMs) for flow cytometry analysis, and viral load was quantified from 140 µL of culture supernatant.

### Total RNA extraction and Nanopore cDNA library preparation
Total RNA was isolated from primary human monocytes using a RNeasy Micro Kit (Qiagen), following the manufacturer's instructions. The RNA samples were further treated with DNase I (Qiagen) and cleaned up using a RNeasy MinElute Cleanup Kit (Qiagen). cDNA libraries were generated from a total of 50 ng RNA according to the Oxford Nanopore Technologies (Oxford Nanopore Technologies Ltd, Oxford, UK) protocol "cDNA-PCR Barcoding Sequencing" with 18 cycles of PCR (4.5 min elongation time). ONT adapters were ligated to the amplified cDNA library.

### Nanopore sequencing and differential gene expression analysis
Nanopore libraries were sequenced using a MinION Mk1B with R9.4.1 (PCS108 LR and RNA001 LR) or R9.5 flow cells (TELO LR). The data were generated using MinKNOW 1.11.5 and base called with Guppy GPU base caller. Debarcoding was performed using the SQK-PBK004 Barcoding Kit (Oxford Nanopore Technologies). STAR aligner[60] was used to map paired-end raw reads to human genome build GRCh38 and counted for genes using featureCounts[61] based on GENCODE v31 gene annotation[62]. Log$_2$-transformed counts per million mapped reads (log$_2$CPM), and log$_2$-transformed reads per kilobase per million mapped reads (log$_2$RPKM) were computed using the edgeR Bioconductor package[63]. Data are accessible at NCBI's Gene Expression Omnibus (GEO) database (GSE291153). Genes with log$_2$CPM inter-quartile range (IQR) of less than 0.5 across all samples were filtered out from subsequent differentially expressed gene (DEG) analysis. DEG analysis was done using edgeR[63].

### Flow cytometry for ex vivo experiments
Following harvesting, human monocytes and MDMs were specifically stained for the surface markers: CD45, CD14, CCR2, HLA-DR, CD16, CD11b, and CD169 (for monocytes); or CD45, CD14, HLA-DR, CD64, CD86, CD163, CD206, and CD169 (for MDMs). Dead cells were excluded by staining with the LIVE/DEAD Fixable Aqua Dead Cell Stain Kit (Life Technologies). The stained cells were subsequently incubated with FACS lysing solution (BD Biosciences) to lyse the red blood cells.

Data were acquired on a Fortessa flow cytometer (BD Biosciences) with BD FACSDiva software version 9.0. Data analysis was performed using FlowJo version 9.3.2 (Tree Star, Inc.).

## Mice

Female wild-type (WT) C57BL/6J mice, aged 4 weeks, were bred and maintained under specific pathogen-free conditions at the Biological Resource Center (BRC) of the Agency for Science, Technology, and Research, Singapore (A*STAR). All experimental procedures involving mice were approved by the Institutional Animal Care and Use Committee (IACUC 241883) of A*STAR, and in compliance with the guidelines of the Agri-Food and Veterinary Authority (AVA) and the National Advisory Committee for Laboratory Animal Research of Singapore (NACLAR). All animal experiments were conducted in an Animal Biosafety Level 2 (ABSL-2) facility.

## Tissue total RNA isolation

For total RNA extraction from tissues, mice were anesthetized with ketamine (150 mg/kg) and xylazine (10 mg/kg), followed by perfusion with PBS. Joint footpads (including the ankle joint and footpad) and spleens were collected and stored in TRIzol reagent (Invitrogen) at −80 °C. Tissues were homogenized using a rotor-stator homogenizer (Xiril Dispomix) at 4000 rpm for 15 s. The homogenized samples were mixed with 230 μL of chloroform and centrifuged at 12,000 rpm for 10 min at 4 °C. The aqueous phase was collected and used for total RNA isolation with the RNeasy Mini Kit (Qiagen), following the manufacturer's instructions. The concentration of extracted total RNA was measured using a NanoDrop 1000 spectrophotometer (Thermo Scientific).

## Gene expression study

Total RNA was extracted from harvested cells using the RNeasy Mini Kit (Qiagen), following the manufacturer's instructions. Quantitative real-time PCR (qRT-PCR) was performed using the QuantiNova SYBR Green PCR Kit (Qiagen) in a 12.5 μL reaction volume, according to the manufacturer's recommendations. All reactions were run on a Bio-Rad CFX384 Real-Time PCR System. Thermal cycling conditions were as follows: 50 °C for 10 min, 95 °C for 2 min, followed by 40 cycles of 95 °C for 5 s and 60 °C for 10 s. Fold changes in gene expression were calculated relative to the uninfected control group and presented using the $2^{-\Delta\Delta CT}$ method.

## Viral RNA extraction and viral load analysis

Viral RNA was extracted from the culture supernatant using a QIAamp Viral RNA Mini Kit (Qiagen), following the manufacturer's instructions. TaqMan probe quantitative real-time PCR (qRT-PCR)[64] was performed to quantify CHIKV NS1 RNA using a QuantiTect Probe RT-PCR kit (Qiagen).

## Western blotting

To detect the phosphorylation status of PI3K/Akt kinases following sialokinin stimulation, cells were lysed with lysis buffer (Cell Signaling Technology) containing 2 mM PMSF. Lysates were separated by SDS-PAGE and subjected to western blotting using the following antibodies: P-PI3K (p85; Tyr458/p55; Tyr199, Cell Signaling Technology, 1:2000), PI3K (p85, Cell Signaling Technology, 1:2000), P-Akt (Ser473, Cell Signaling Technology, 1:2000), and Akt (Cell Signaling Technology, 1:2000). Signal intensities were visualized using a ChemiDoc imaging system (Bio-Rad) and quantified using Image Lab software version 6.0.1 (Bio-Rad).

## In vivo virus infection

Mice were subcutaneously inoculated with $1 \times 10^6$ pfu of CHIKV in 30 μL PBS at the ventral side of the right hind footpad. Disease progression in terms of joint swelling and viremia was monitored daily until 10 dpi, as previously described[65]. Joint swelling was determined by measuring the height and width of the footpad with an electronic calliper (Mitutoyo) and calculated as the relative increase compared to pre-infection (0 dpi) with the following formula: [(x−day 0)/day 0], where x is the footpad size measurement for a given dpi. To track viremia, 10 μL of blood from the tail vein were collected and mixed in 10 μL of citrate-phosphate-dextrose solution (Sigma-Aldrich) with 120 μL PBS.

## Mouse spleen and joint cell isolation

Mice were infected with CHIKV and euthanized by cervical dislocation at 6 dpi. The footpads and ankles were removed, de-skinned, and placed immediately in a 4 mL digestion medium containing dispase (2 U/mL; Invitrogen), collagenase IV (20 μg/mL; Sigma-Aldrich), and DNase I mix (50 μg/mL; Roche Applied Science) in complete Roswell Park Memorial Institute (RPMI) medium. Spleens were dissociated in RPMI 1640 medium containing 10% FBS (complete RPMI). The cells were then isolated as previously described[40].

## Phenotyping of leukocytes

Isolated joint cells were first blocked with 1% mouse/rat serum (Sigma-Aldrich) blocking buffer for 10 min. The cells were then stained for 20 min with the following antibodies: BUV395-conjugated anti-mouse CD45 (clone 30-F11; BD Biosciences), Pacific Blue-conjugated anti-mouse CD4 (clone RM4-5; BioLegend), CF594-conjugated anti-mouse CD8 (clone 53-6.7; BD Biosciences), PE-Cy7-conjugated anti-mouse CD3 (clone 17A2; BioLegend), APC-Cy7-conjugated anti-mouse Ly6C (clone HK1.4; BioLegend), Alexa Fluor 700-conjugated anti-mouse MHC-II (clone M5/114.15.2; BioLegend), PerCP-Cy5.5-conjugated anti-mouse LFA-1 (clone H155-78; BioLegend), BV650-conjugated anti-mouse CD11b (clone M1/70; BioLegend), BV605-conjugated anti-mouse CD11c (clone N418; BioLegend), CF594-conjugated anti-mouse Ly6G (clone 1A8; BD Biosciences), eFluor450-conjugated anti-mouse B220 (clone RA3-6B2; eBioscience), PE-conjugated anti-mouse MerTK (clone DS5MMER; eBioscience), APC-conjugated anti-mouse CD64 (clone X54-5/7.1; BioLegend), biotin-conjugated anti-mouse NK1.1 (clone PK136; eBioscience), and BUV737 streptavidin (BD Biosciences). Samples were acquired on a LSR II flow cytometer (BD Biosciences) with FACSDiva software and analysed using FlowJo software.

## T cell IFN-γ ELISpot assay

The ELISpot assay was performed according to the manufacturer's instructions (Mabtech) using the ELISpot Plus: Mouse IFN-γ (ALP) kit. Mice were sacrificed at 6 days post-infection (dpi), and joint footpad tissues and spleens were harvested and processed as described above. Ex vivo CD4+ T cell enrichment was carried out using the Mouse CD4+ T Cell Isolation Kit II (Miltenyi Biotec), following the manufacturer's protocol. CHIKV-specific T cell stimulation was conducted in complete RPMI medium supplemented with 30 U/ml IL-2 and $1.5 \times 10^6$ CHIKV virions per well, using 5000 isolated T cells. Complete RPMI containing 30 U/ml IL-2 was used as a negative control. For all wells containing joint footpad cells, $1.5 \times 10^5$ splenocytes from non-infected mice were added as antigen-presenting cells (APCs). Plates were incubated at 37 °C with 5% $CO_2$ for 15 h. After incubation, cells were removed, and wells were developed according to the manufacturer's instructions. Spot counting and imaging were performed using the MABTECH IRIS system.

## Cytokine profiling in the joints

Mice were anesthetized by intraperitoneal injection of 150 mg of ketamine and 10 mg/kg xylazine cocktail, followed by intracardial perfusion with PBS at the indicated time points. Subsequently, the footpads were collected and homogenized in 1.5 mL RIPA buffer (50 mM Tris-HCl pH 7.4; 1% NP-40; 0.25% sodium deoxycholate; 150 mM NaCl; 1 mM EDTA) with 1× protease inhibitors (Roche) using a

gentleMACS M tube with a gentleMACS Dissociator (Miltenyi Biotec). Cell lysates were then sonicated at 70% intensity for 15 s (Branson Ultrasonics Sonifier™ S-450), and the supernatants were collected for cytokine/chemokine quantification as described below. Data are expressed as pg/mL in footpad lysate.

### Multiplex microbead immunoassay for cytokine quantification

Cytokine and chemokine concentrations in mice serum and footpad lysates were quantified simultaneously using a multiplex microbead-based immunoassay, the Cytokine & Chemokine 36-Plex Mouse ProcartaPlex Panel 1 A (EPX360-26092-901; Thermo Scientific), following the manufacturer's protocol. Data were acquired using a Luminex FlexMap 3D instrument (Millipore) and analysed with Bio-Plex Manager™ 6.0 software (Bio-Rad) based on standard curves plotted through a five-parameter logistic curve setting. The cytokines and chemokines assayed included: IFN-γ, IL-12p70, IL-13, IL-1β, IL-2, IL-4, IL-5, IL-6, TNF-α, GM-CSF, IL-18, IL-10, IL-17A, IL-22, IL-23, IL-27, IL-9, GRO-α, IP-10, MCP-1, MCP-3, MIP-1α, MIP-1β, MIP-2, RANTES, eotaxin, IFN-α, IL-15/IL-15R, IL-28, IL-31, IL-1α, IL-3, G-CSF, LIF, ENA-78/CXCL5, and M-CSF.

### Evaluation of the level of antibodies against sialokinin

To test the level of antibodies against sialokinin, 96-well ELISA plates (Nunc Maxisorp) were coated with 50 µL/well of sialokinin peptides (2 µg/mL) prepared in PBS and were incubated overnight at 4 °C. Plates were then blocked for 1.5 h with 5% milk powder in 1× PBST (blocking buffer) at 37 °C and incubated with 100 µL/well of a 1/100 dilution of human sera in sample diluent (2.5% milk powder in 1× PBST) at 37 °C for 1 h. The plates were then incubated with 100 µL/well of horseradish peroxidase (HRP)-labeled goat anti-human IgG (1:1000) antibodies (Jackson ImmunoResearch) at 37 °C for 30 min. Readout was detected with TMB substrate solution (Sigma-Aldrich) and terminated with sulfuric acid (Sigma-Aldrich). Absorbance was measured at 450 nm in a microplate autoreader (Tecan). Peptides are considered positive if absorbance values are at least twofold higher than the mean values of pooled healthy donor samples.

### Statistical analysis

Statistical analyses were performed using GraphPad Prism (version 9.0.0; GraphPad Software). Parametric or non-parametric tests were used to compare between groups and controls, as appropriate. $p$-values less than 0.05 were considered statistically significant. Plots were generated using GraphPad Prism version 9.0.0.

### Reporting summary

Further information on research design is available in the Nature Portfolio Reporting Summary linked to this article.

## Data availability

All data supporting the findings of this study are available within the paper, its Supplementary information or Source data files. The complete RNA sequencing dataset was deposited in the National Center for Biotechnology Information's Gene Expression Omnibus Database and is accessible through GSE291153. Source data are provided with this paper.

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

## Acknowledgements

The authors would like to thank the participants who donated blood samples to this study. The authors are grateful to the A*STAR IDL Pathogen Flow Platform for their invaluable assistance in this study. The authors are also grateful to Dr. Olaf Rötzschke, Wilson How, and Norman Leo Fernandez from the Singapore Immunology Network (SIgN) Multiplex Analysis of Proteins (MAP) platform for their assistance in running multiplex microbead-based immunoassay. We thank the study participants and healthy volunteers for their participation in the study and the clinical and research staff from the Communicable Disease Centre/Tan Tock Seng Hospital for assistance in patient enrollment, study coordination, data entry, and blood sample preparation. We thank Dr. Daniel Ackerman of Insight Editing London for editing the manuscript prior to submission. The work was supported by the Ministry of Education, Singapore (MOE2015-T3-1-003)(RMK, LFPN), the National Medical Research Council (OF-LCG19May-0034)(LFPN), core research grants provided by the Biomedical Research Council (BMRC) from the Agency for Science,

Technology and Research (A*STAR)(SWF, LFPN), as well as a Career Development Fund (C210812043) and Central Research Fund (CRF) awarded to SWF by A*STAR. SWF was supported by MOE AcRF Tier 3 grant (MOE2015-T3-1-003) from 2016 to 2020. The funders had no role in the design and conduct of the study; collection, management, analysis, and interpretation of the data; preparation, review, or approval of the manuscript; and decision to submit the manuscript for publication.

## Author contributions

S.W.F. conceptualized the study, acquired, analyzed, interpreted the data, and wrote the manuscript. J.J.L.T., V.S., S.N.A., V.K.X.N., N.W., Y.H.C., A.T.R., L.H.L., A.X.Y.L., S.K.W.T., R.S.L.C., T.K.C., A.R., G.C., F.M.L., and L.R. acquired, analyzed, interpreted the data, and reviewed the manuscript. B.L. analyzed the Nanopore sequencing data. Y.S.L. designed the clinical study protocol and collected the clinical data. R.M.K. and L.F.P.N. conceptualized the study and reviewed the manuscript. All authors approved the final version of the manuscript.

## Competing interests

A captioned provisional patent application has been filed (Singapore Provisional Application No. 10202502148Y): Sialokinin antibodies as biomarkers for prognostic tools in arbovirus infections (S.W.F., A.X.Y.L., L.R., R.M.K., and L.F.P.N.). The remaining authors declare no conflicts of interest.

## Additional information

[1]A*STAR Infectious Diseases Labs (A*IDL), Agency for Science, Technology and Research (A*STAR), Singapore, Singapore. [2]Life Technologies Holdings Pte Ltd, Singapore, Singapore. [3]Department of Biological Sciences, National University of Singapore, Singapore, Singapore. [4]Lee Kong Chian School of Medicine, Nanyang Technological University, Singapore, Singapore. [5]Environmental Health Institute, National Environment Agency, Singapore, Singapore. [6]Infectious Diseases Translational Research Programme, Yong Loo Lin School of Medicine, National University of Singapore, Singapore, Singapore. [7]National Centre for Infectious Diseases, Singapore, Singapore. [8]Saw Swee Hock School of Public Health, National University of Singapore, Singapore, Singapore. [9]School of Biological Sciences, Nanyang Technological University, Singapore, Singapore. [10]Department of Pharmacology, Yong Loo Lin School of Medicine, National University of Singapore, Singapore, Singapore. [11]Department of Biochemistry and Molecular Biology, Virginia Commonwealth University, Richmond, VA, USA. [12]Department of Biochemistry, Yong Loo Lin School of Medicine, National University of Singapore, Singapore, Singapore. ✉e-mail: fong_siew_wai@a-star.edu.sg; lisa_ng@a-star.edu.sg

