## [Transparent Peer Review file · Nature Communications]

Mosquito salivary sialokinin reduces monocyte activation and chikungunya virus-induced inflammation via neurokinin receptors

Corresponding Author: Dr Siew-Wai Fong

Version 0:

Reviewer comments:

Reviewer #1

(Remarks to the Author)

The manuscript by Siew-Wai Fong et al. entitled "Mosquito salivary sialokinin reduces monocytes activation and chikungunya-induced inflammation via neurokinin receptors" explores the role of *Aedes aegypti* sialokinin in the modulation of host immunity during chikungunya virus infection. The authors demonstrate that sialokinin modulates the activity of monocytes including the downregulation of CD169. They propose that this is mediated through the signaling of sialokinin on the NK1 and 2 receptors resulting in the activation of PI3K and Akt. They further show that the administration of sialokinin on the day of infection results in decreased joint inflammation including a reduction in the number of CD4+IFN γ + T cell infiltrating into the foot. A similar reduction in inflammation was also observed with sialokinin was administered a few days after infection. Finally the authors provide evidence that in a cohort of acutely CHIKV infected patients that these patients demonstrate an elevated IgG antibody response to sialokinin which positively correlates with disease severity, viral loads, and CRP. Overall, this is an interesting body of work exploring the immunomodulatory role of a mosquito salivary gland peptide. While the authors provide evidence that sialokinin can signal through neurokinin receptors on monocytes in vitro, there are limitations to the interpretation of these results as outlined below. In addition, there is no data provided in vivo to demonstrate that these are the receptors mediating the CHIKV induced inflammation. This concern as well as a few others diminish enthusiasm for this paper.

1. In figure 1F the authors nicely demonstrate the reduction in expression of CD169 induced by sialokinin on monocytes is dependent upon p13K signaling, since the inhibitor LY-294002 completely restored CD169 expression. However, the NK1R and NK2R inhibitors each only had modest effects leaving open the possibility that sialokinin is signaling through a receptor(s) other than the NK receptors. Stronger evidence should be provided that this is the only receptor that sialokinin is signaling through. For example, the authors could evaluate the impact of combining CP-96345 and GR-159897 on blocking the downregulation of CD169 that is induced by sialokinin.

2. The data presented in figure 2, while statistically significant, is rather underwhelming. It is also unclear why in panel A the histograms are overlaid while in panel b they are presented separately. By eye, the histograms do not look different between the CHIKV only and CHIKV + sialokinin group. If the authors feel there is a difference in CHIKV infectivity they may need to look at alternative time points after infection or consider growth curves to measure replicating virus. Alternatively, it may be interesting to determine the impact of pretreating cells with sialokinin and then looking at the impact of infecting with CHIKV.

3. While the differences seen in viremia are interesting, a more relevant tissue to measure viral titers in would be the foot tissue. This is where sialokinin is being injected in figure 3. In addition, the analysis of cellular infiltrates and cytokine/chemokine levels in the foot tissues in both figure 3 and 4 are difficult to interpret without knowing what viral titers look like in this tissue.

4. As noted by the authors in the discussion (lines 321-322), sialokinin functions to enhance spread of the virus. The authors should determine if this treatment results in increased viral titers in tissues outside of the injected foot (muscle, contralateral foot, spleen).

5. The most striking finding in the cellular infiltrate data in the decrease in the number of CD4+IFN γ + T cells with sialokinin treatment. However, these cells were stimulated with PMA/Ion and do not necessarily represent antigen specific T cells. This data and the model proposed by the authors would be much stronger if this analysis was done looking at Ag specific T cells such as with an Elispot for CHIKV stimulated IFN γ producing cells.

6. It has been previously reported that sialokinin induces vascular leak. The authors in figure 3b show a reduction in CD169+ monocytes in the blood of infected mice. What happens to the numbers of other cell populations within the blood?
7. The data in the extended data figure 5C and D should be shown in the main figure (Fig. 3)
8. In figure 1d, for the panel showing the quantification of P-PI3K/PI3K and P-Akt/Akt please clarify what time point this represents (20, 40, or 60 minutes post stimulation). Also please clarify what is being compared in the statistical analysis for this figure.

Reviewer #2

(Remarks to the Author)

The manuscript of Fong and co-authors explores the role of a mosquito salivary peptide, sialokinin, in modulating host immune responses to chikungunya virus (CHIKV) infection. The study finds that sialokinin binds to human neurokinin receptors NK1R and NK2R and suppresses activation of monocytes and macrophages, which are key drivers of CHIKV-induced inflammation. Authors show that this immunomodulatory effect is mediated via the PI3K/Akt signaling pathway, leading to reduced expression of inflammatory markers like and diminished production of IFN- γ by CD4+ T cells. In mouse models, sialokinin treatment reduced joint swelling, viremia, and pro-inflammatory cytokine levels. Authors speculate that sialokinin could be used as a therapeutic target to mitigate viral-induced inflammation in CHIKV and potentially other mosquito-borne diseases. This overall concept of this study is compelling, however it contains major flaws in statistical analyses and data interpretation, which make me doubt the conclusions of the authors. My major concerns are outlined below.

1. The significance and fold-change for the differentially expressed genes in Fig 1A is unclear as the volcano plot lacks scale. In addition, authors observe down-regulation of the IFN-stimulated genes (e.g. IFIT2) in sialokinin-treated uninfected cells (line 159). However, ISGs are normally not expressed or expressed at marginal levels. Authors should provide a supplementary table with normalized and raw read counts for all DEGs in this analysis to show whether the expression levels and changes observed are biologically meaningful. Authors should repeat this experiment with CHIKV-infected monocytes to show whether decreased activation is observed in the relevant infection context.
2. How many replicates were used to calculate Pearson correlation coefficient in Fig 1c? These coefficients are not informative for small sample sizes.
3. Patterns of data points in the groups shown in Fig 1d, 1f, Fig 2, Fig 4c,d, don't follow normal distribution. Therefore, non-parametric statistical tests must be used instead of t-test, as authors correctly did for Fig 5b.
4. Data normalization and statistical analyses in Fig 1a and 2 are conceptually flawed as all treated samples have been normalized against control and then compared to it (with the control group having variance of zero). This highly distorts the data unless all samples in the control group had exactly the same value. In addition, it made data from different groups dependent, while all statistical tests assume independence of data points.
5. Considering that virus replicates logarithmically, data on viral RNA copies in Fig 2b must be log₁₀ transformed prior to statistical comparison. Values must be plotted on logarithmic scale. What is the reason for not showing individual data point on viral copy number plots. What were the viral titers in the culture fluids? Additionally, the selection of the time point for this analysis is not justified. Authors should determine viral growth kinetics through multiple time points.
6. Data in Fig3C and D is conflicting with 4B, as on one hand treatment reduces viral load and inflammation while in figure 4B there is no effect on viral load and mild effects on inflammation. This suggests the sialokinin may be affecting viral replication and the reduced inflammation may simply result from reduced viral load. This should be discussed in the paper.

Reviewer #3

(Remarks to the Author)

The manuscript by Fong et al. explores the immunomodulatory effects of the *Aedes aegypti* salivary peptide sialokinin during Chikungunya virus (CHIKV) infection. It is a novel and impactful study where the authors demonstrate that sialokinin binds to neurokinin receptors on monocytes and limits their activation, evidenced by downregulation of several type-I interferon stimulated genes (ISGs) and surface activation markers, as well as their infectivity. The study links these findings to downstream inflammation and T cell activation both in vitro and in vivo, and further correlates anti-sialokinin IgG titers with disease severity and inflammation in CHIKV patients.

The manuscript is well executed and addresses an important aspect in our understanding of vector-host-virus immunological dynamics. The integration of transcriptomics, functional assays, in vivo models and patient samples is advantageous and exciting. The mechanistic insights into neurokinin receptor signaling are novel and presented with clarity and overall, the findings are robust and of potential translational relevance.

Below are minor comments:

- The authors cite prior work (Reference 32) showing that mosquito saliva enhances infection through sialokinin-dependent vascular leakage. A brief discussion of how their findings reconcile with this study would be helpful and could contextualize the multifaceted role of sialokinin during CHIKV infection.
- The authors should consider measuring phosphorylation of PI3K and AKT in PBMCs derived from mice treated with sialokinin to strengthen the hypothesis that this is the relevant signaling pathway activated.
- The human patient cohort seems to have a sex bias (Out of 30 individuals, 26 are male vs. 4 female). The authors should discuss how this may impact interpretation.
- In line 60, the sentence "in a warming sicker world" is impactful but rephrasing it could be considered.

- In the in vitro experiments (Figures 1 and 2) it is not clear for how long the incubation with sialokinin was.
- The figure legends should include an indication how many experiments were conducted.

Reviewer #4

(Remarks to the Author)

Reviewer #5

(Remarks to the Author)

Version 1:

Reviewer comments:

Reviewer #1

(Remarks to the Author)

This is a revised manuscript by the authors. They have done a very nice job of addressing the concerns raised by the reviewers. In addition, several pieces of new data including the studies in which both receptors were blocked and the studies exploring the antigen specific responses of infiltrating T cells during sialokinin blockade have strengthened their conclusions.

Reviewer #2

(Remarks to the Author)

The authors have addressed all my concerns.

Reviewer #3

(Remarks to the Author)

My comments have been nicely addressed, and I am pleased to see the improvements introduced in the revised manuscript.

Reviewer #4

(Remarks to the Author)

Reviewer #5

(Remarks to the Author)

13 Jul 2025

A point-by-point response to reviewers' comments for Nature Communications Submission NCOMMS-25-08509-T

REVIEWER COMMENTS

Reviewer #1 (Remarks to the Author):

The manuscript by Siew-Wai Fong et al. entitled “Mosquito salivary sialokinin reduces monocytes activation and chikungunya-induced inflammation via neurokinin receptors” explores the role of *Aedes aegypti* sialokinin in the modulation of host immunity during chikungunya virus infection. The authors demonstrate that sialokinin modulates the activity of monocytes including the downregulation of CD169. They propose that this is mediated through the signaling of sialokinin on the NK1 and 2 receptors resulting in the activation of PI3K and Akt. They further show that the administration of sialokinin on the day of infection results in decreased joint inflammation including a reduction in the number of CD4+IFN γ + T cell infiltrating into the foot. A similar reduction in inflammation was also observed with sialokinin was administered a few days after infection. Finally the authors provide evidence that in a cohort of acutely CHIKV infected patients that these patients demonstrate an elevated IgG antibody response to sialokinin which positively correlates with disease severity, viral loads, and CRP. Overall, this is an interesting body of work exploring the immunomodulatory role of a mosquito salivary gland peptide. While the authors provide evidence that sialokinin can signal through neurokinin receptors on monocytes *in vitro*, there are limitations to the interpretation of these results as outlined below. In addition, there is no data provided *in vivo* to demonstrate that these are the receptors mediating the CHIKV induced inflammation. This concern as well as a few others diminish enthusiasm for this paper.

Response: We thank the reviewers for their thorough evaluation and encouraging assessment of our work. We appreciate the recognition of our findings regarding the immunomodulatory role of sialokinin during CHIKV infection. In our *in vitro* study, we identified NK₁R and NK₂R as the primary neurokinin receptors mediating the effects of sialokinin on monocyte activation and immune modulation. These insights provide a strong mechanistic foundation in understanding how sialokinin influences host immunity. While direct *in vivo* demonstration of receptor involvement was beyond the scope of the current study, we have now explicitly discussed this point in the revised Discussion section. We agree that future studies employing NK₁R and NK₂R double knockout mouse models may be valuable for further exploring the therapeutic potential of these neurokinin receptors in CHIKV-induced inflammation and other immune-mediated diseases. We also addressed the reviewer's specific concerns regarding data interpretation and provided clarifications and revisions where appropriate in the responses below.

The revised Discussion on page 21, lines 457 to 465 reads:

‘Our *in vitro* data demonstrate that both NK₁R and NK₂R serve as primary receptors mediating the immunomodulatory effects of sialokinin, acting through PI3K/Akt signaling to suppress monocyte activation. This discovery provides an avenue in targeting the NK receptor or its downstream signaling pathway to ameliorate CD169+ macrophage-mediated inflammation. Future studies employing double knockout mouse models for NK1R and NK2R will be valuable for exploring the therapeutic potential of sialokinin and its derivatives in treating chronic inflammatory diseases, including chikungunya’.

1. In figure 1F the authors nicely demonstrate the reduction in expression of CD169 induced by sialokinin on monocytes is dependent upon pi3K signaling, since the inhibitor LY-294002

completely restored CD169 expression. However, the NK1R and NK2R inhibitors each only had modest effects leaving open the possibility that sialokinin is signaling through a receptor(s) other than the NK receptors. Stronger evidence should be provided that this is the only receptor that sialokinin is signaling through. For example, the authors could evaluate the impact of combining CP-96345 and GR-159897 on blocking the downregulation of CD169 that is induced by sialokinin.

Response: We thank the reviewer for this thoughtful and constructive comment. Specifically, the suggestion to explore dual receptor blockade to further strengthen our conclusions regarding receptor specificity.

In response, we have performed additional experiments in which monocytes were pre-treated with both the NK₁R inhibitor CP-96345 and the NK₂R inhibitor GR-159897 prior to sialokinin stimulation. As suggested, we further assessed whether simultaneous inhibition of both receptors would enhance the suppression of CD169 downregulation.

Our new results (presented in Figure 1f) demonstrate that combined inhibition of NK₁R and NK₂R leads to complete restoration of CD169 expression to baseline levels, comparable to that observed with the PI3K inhibitor LY-294002. This suggests that PI3K signaling functions downstream of NK1R and NK2R activation, and that these two receptors mediate the dominant pathway through which sialokinin acts on monocytes.

This additional data significantly strengthens our conclusion that NK₁R and NK₂R are the primary receptors responsible for sialokinin-induced CD169 downregulation. While we do not entirely exclude the possibility of minor contributions from other pathways, the dual inhibition results provide compelling evidence for the central role of these neurokinin receptors. We have updated the Results sections to include these new data.

Results on page 10, lines 201 to 215 now read “We subsequently examined the downstream signaling linkage between the NK receptor/PI3K/Akt pathway and sialokinin-mediated regulation of monocyte activation by pre-treating cells with NK receptor antagonists (CP-96345 or GR-159897), or with the PI3K inhibitor LY-294002 (Fig. 1e). Individually, the NK receptor antagonists only partially neutralized the sialokinin-mediated downregulation of Siglec-1/CD169 expression in human monocytes, while the PI3K inhibitor completely blocked this downregulation (Fig. 1f). To further investigate the contribution of NK₁R and NK₂R, additional experiments were performed where monocytes were pre-treated with both CP-96345 and GR-159897 prior to sialokinin stimulation. As shown in Figure 1f, dual receptor blockade resulted in a complete restoration of CD169 expression to baseline levels, comparable to PI3K inhibition. These findings suggest that PI3K signaling functions downstream of NK₁R and NK₂R activation, and that both receptors are required for sialokinin-mediated modulation of monocyte activation’.

Fig 1 f, Treatment with NK receptor antagonists or a PI3K inhibitor inhibits sialokinin-mediated PI3K/Akt activation and restores CD169 expression in human monocytes. Freshly purified human monocytes were incubated with or without sialokinin peptides. The monocytes were pre-treated with the NK1R antagonist (CP-96345; n = 5), NK2R antagonist (GR-159897; n = 5), both antagonists in combination (n = 4), or the PI3K inhibitor (LY-294002; n = 5) prior to sialokinin stimulation. Cells were harvested 24 hours post-treatment for analysis of activation by flow cytometry. CD169 expression levels are presented as mean fluorescence intensity (MFI) and also expressed as fold change relative to non-treated controls. Individual inhibition of NK1R or NK2R partially blocked sialokinin-induced downregulation of CD169, while dual receptor blockade fully restored CD169 expression to baseline levels, comparable to PI3K inhibition. Statistical comparisons between treatment groups and controls were performed using paired two-tailed t-test or Wilcoxon signed-rank test, depending on data distribution. $p < 0.05$, $p < 0.01$ indicate statistically significant differences.

2. The data presented in figure 2, while statistically significant, is rather underwhelming. It is also unclear why in panel A the histograms are overlaid while in panel b they are presented separately. By eye, the histograms do not look different between the CHIKV only and CHIKV + sialokinin group. If the authors feel there is a difference in CHIKV infectivity they may need to look at alternative time points after infection or consider growth curves to measure replicating virus. Alternatively, it may be interesting to determine the impact of pretreating cells with sialokinin and then looking at the impact of infecting with CHIKV.

Response: We thank the reviewer for this thoughtful feedback and constructive suggestion. Regarding the presentation of histograms in Figure 2, we would like to clarify that Panel A overlays the histograms to facilitate comparison of mean fluorescence intensity (MFI), which in this case reflects the expression level of CD169 on the cell surface. This format enables clearer visualization of shifts in CD169 expression between the CHIKV-only and CHIKV + sialokinin groups. In contrast, Panel B presents the histograms separately to highlight the percentage of CHIKV-infected cells, which is a distinct and complementary parameter. This dual approach was chosen to illustrate both the phenotypic modulation of host cells and the extent of viral infection.

To clarify, we did perform kinetic experiments at 24- and 48-hours post-infection (hpi) to determine the optimal time points for assessing CHIKV infectivity and CD169 expression on both human monocytes and monocyte-derived macrophages (MDMs). Our results indicate that 24 hpi is optimal for monocytes, while 48 hpi is optimal for MDMs. These time points were selected based on the observed differences in both infectivity and CD169 expression. We have now included these data in the Extended Data Figures 5 & 6 and updated the Results section on page 11, lines 225 to 232 to read ‘To determine the optimal time points for assessing CHIKV infectivity and CD169 expression following sialokinin treatment, kinetic experiments were performed at 24- and 48-hours post-infection (hpi) using both human monocytes and monocyte-derived macrophages (MDMs). These experiments revealed that 24 hpi is optimal for monocytes, while 48 hpi is optimal for MDMs, based on the observed differences in magnitude of infection and CD169 expression levels with sialokinin treatment (Extended Data Figs. 5 & 6). These time points were subsequently used in all downstream analyses’.

Extended Data Fig. 5: Effect of sialokinin on human monocyte activation and CHIKV infection at 24- and 48-hours post-infection. Human monocytes from healthy donor were infected with CHIKV in the presence or absence of sialokinin (10 μ M). Activation profiles of monocytes were assessed by flow cytometry. CD169 expression was lower in sialokinin-treated cells following CHIKV infection (MOI = 10) at 24 hours post-infection. CHIKV infectivity in monocytes was reduced in the presence of sialokinin.

Extended Data Fig. 6: Effect of sialokinin on human monocyte-derived macrophages (MDMs) activation and CHIKV infection at 24 and 48 hours post-infection. Human MDMs from healthy donor were infected with CHIKV in the presence or absence of sialokinin (10 μ M). Activation profiles of monocytes were assessed by flow cytometry. CD169 expression was lower in sialokinin-treated cells following CHIKV infection (MOI = 10) at 24 hours post-infection. CHIKV infectivity in MDMs was reduced in the presence of sialokinin.

To further address the reviewer's concerns, we conducted additional experiments in which human monocytes were pre-treated with sialokinin prior to CHIKV infection. Interestingly, we observed that pre-treatment with sialokinin led to a reduction in both CHIKV infection and CD169 expression. These findings suggest that sialokinin exerts a modulatory effect on monocyte susceptibility to CHIKV, potentially by altering the activation state of the cells. This result complements our initial observations and provides further evidence supporting the role of sialokinin in shaping host cell responses to viral infection. We have now included these data in the Extended Data Figure 8 and updated the Results section on page 12, lines 248 to 252 to read 'To further investigate the role of sialokinin, we conducted pre-treatment experiments in which human monocytes were exposed to sialokinin prior to CHIKV infection. Notably, pre-treatment with sialokinin reduced both CHIKV infection and CD169 expression (Extended Data Fig. 8), further supporting the role of sialokinin in modulating monocyte susceptibility to CHIKV'.

Extended Data Fig. 8: Pre-treatment of human monocytes with sialokinin reduces CHIKV infection and CD169 expression at 24 hours post-infection. Human monocytes isolated from healthy donors were pre-treated with sialokinin (10 μ M) for 1 hour prior to infection with CHIKV (MOI = 10). At 24 hours post-infection, cells were analyzed by flow cytometry to assess activation profiles and infection levels. Pre-treatment with sialokinin resulted in a marked reduction in CD169 surface expression and CHIKV

infectivity compared to untreated controls. These findings suggest that sialokinin modulates monocyte susceptibility to CHIKV by dampening both viral infection and activation marker expression. * $p < 0.05$, ** $p < 0.01$ by paired two-tailed t-test.

3. While the differences seen in viremia are interesting, a more relevant tissue to measure viral titers in would be the foot tissue. This is where sialokinin is being injected in figure 3. In addition, the analysis of cellular infiltrates and cytokine/chemokine levels in the foot tissues in both figure 3 and 4 are difficult to interpret without knowing what viral titers look like in this tissue.

Response: We thank the reviewer for highlighting the importance of assessing viral titers in the footpad tissue, particularly in the context of sialokinin injection. To address this, we performed additional experiments to quantify viral titers in the footpad tissue at 6 days post-infection (dpi). Specifically, we measured viral loads in the right footpad, which corresponds to the site of sialokinin and virus injection as well as the immune profiling site for both Figures 3 and 4. Our results showed no significant difference in viral titers between the CHIKV-only and CHIKV + sialokinin groups at this time point. These findings suggest that the local immune modulation observed in Figures 3 and 4 is not driven by sustained differences in viral burden at 6 dpi.

We have now included these data in the updated Figures 3e and 4c, updated the Results section on page 14, lines 297 to 302 to read 'We hypothesized that differences in joint inflammation may be attributable to differences in the profile of leukocytes infiltrating the joints. To investigate this, we first measured viral titers in the joint footpad tissue at 6 dpi and found no significant difference in viral load between the CHIKV-only and CHIKV + sialokinin groups (Fig. 3e), suggesting that the observed immune modulation is not due to differences in local viral replication' and also page 15, lines 338 to 342 to read 'To further investigate the local immune response, we measured viral titers in the joint footpad tissue at 6 dpi and found no significant difference in viral load between the treated and untreated groups (Fig. 4c), indicating that the reduced inflammation was not due to altered viral replication at the site of pathology'.

Fig. 3: e, Tissue viral loads were quantified by qRT-PCR of CHIKV nsP1 viral copies at right joint footpad of CHIKV-infected mice (CHIKV, n = 5; CHIKV + sialokinin, n = 5) at 6 dpi. Statistical comparisons between the two groups were performed using unpaired two-tailed t-test.

Fig. 4: c, Joint swelling was monitored to 14 dpi, viremia was determined from blood collected from the tail vein from 1 to 10 dpi, and viral load was quantified by qRT-PCR of CHIKV nsP1 viral copies. Data are presented as mean \pm SD (n = 5 per group). * p < 0.05, ** p < 0.01, *** p < 0.001 by unpaired two-tailed t -test. Tissue viral loads were quantified by qRT-PCR of CHIKV nsP1 viral copies at right joint footpad of CHIKV-infected mice (CHIKV, n = 5; CHIKV + sialokinin, n = 5) at 6 dpi. Statistical comparisons between the two groups were performed using unpaired two-tailed t -test.

4. As noted by the authors in the discussion (lines 321-322), sialokinin functions to enhance spread of the virus. The authors should determine if this treatment results in increased viral titers in tissues outside of the injected foot (muscle, contralateral foot, spleen).

Response: We thank the reviewer for this important suggestion regarding the potential systemic spread of CHIKV following sialokinin treatment. To address this, we performed additional experiments to measure viral titers in tissues outside of the injected foot, including the contralateral (left) footpad and spleen, at 1-day post-infection (dpi), a time point chosen to capture early viral dissemination. Notably, we observed a significantly higher viral load in the contralateral left footpad of mice treated with sialokinin compared to controls, suggesting that sialokinin facilitates early viral spread beyond the site of inoculation. However, no significant differences in viral titers were detected in the right footpad and spleen at this time point.

These findings support the hypothesis that sialokinin enhances CHIKV dissemination, particularly to anatomically adjacent tissues, and provide further mechanistic insight into its role in promoting viral spread. We have included the new data in the updated Figure 3d and also updated the Results section accordingly. The new section in the Results on page 13, lines 285 to 293 now reads 'To investigate whether sialokinin promotes early viral dissemination beyond the injection site, we measured viral titers in peripheral tissues at 1 dpi, including the contralateral (left) footpad, muscle, and spleen. Notably, we observed a significantly higher viral load in the contralateral footpad of sialokinin-treated mice compared to controls (Fig. 3d), suggesting that sialokinin facilitates early viral spread to distal tissues. However, no significant differences in viral titers were detected in the muscle or spleen at this early time point (Fig 3d). These findings support the role of sialokinin in enhancing early viral dissemination, particularly to anatomically adjacent tissues'.

Fig. 3: d, Tissue viral loads were quantified by qRT-PCR of CHIKV nsP1 viral copies at the contralateral left joint footpad, the right joint footpad and the spleen of CHIKV-infected mice (CHIKV, n = 7; CHIKV + sialokinin, n = 7) at 1 dpi. Statistical comparisons between the two groups were performed using unpaired two-tailed t -test or Mann-Whitney U test, depending on data distribution. * p < 0.05.

5. The most striking finding in the cellular infiltrate data in the decrease in the number of CD4⁺/IFN γ ⁺ T cells with sialokinin treatment. However, these cells were stimulated with PMA/Ion and do not necessarily represent antigen specific T cells. This data and the model proposed by the authors would be much stronger if this analysis was done looking at Ag specific T cells such as with an ELISpot for CHIKV stimulated IFN γ producing cells.

Response: We thank the reviewer for this important observation. We agree that stimulation with PMA/ionomycin, while useful for assessing general cytokine-producing capacity, does not specifically reflect antigen-specific T cell responses. To address this, we have performed additional ELISpot assays to directly assess CHIKV-specific CD4⁺ T cell responses. CD4⁺ T cells were isolated from the joint footpad at 6 days post-infection and stimulated *ex vivo* with inactivated CHIKV virions in the presence of IL-2. This approach allowed us to quantify virus-specific IFN- γ -producing CD4⁺ T cells. Consistent with our intracellular cytokine staining results, we observed a significant reduction in the frequency of CHIKV-specific IFN- γ ⁺CD4⁺ T cells in sialokinin-treated mice compared to controls. These findings strengthen our proposed model by demonstrating that sialokinin not only reduces the general inflammatory potential of CD4⁺ T cells but also specifically dampens virus-specific T cell responses at the joint footpad during the peak of inflammation. The new data have been included in the updated Figures 3g and 4, and the Results and Discussion sections have been revised accordingly.

The updated Results to describe Figure 3g on page 14, lines 313 to 318 now reads ‘To directly assess virus-specific responses, we performed ELISpot assays on CD4⁺ T cells isolated from the joint footpad at 6 dpi. These cells were stimulated *ex vivo* with CHIKV virions in the presence of IL-2 to promote antigen-specific cytokine production. Consistent with the intracellular cytokine staining results, we observed a significant reduction in the frequency of CHIKV-specific IFN- γ -producing CD4⁺ T cells in sialokinin-treated mice (Fig. 3g)’.

Fig. 3: g Representative images of ELISpot wells depicting the number of IFN- γ -producing cells in enriched CD4⁺ T cells from joint footpads at 6 dpi. Quantitative plots show the total numbers of IFN- γ -producing CD4⁺ T cells per infected joint footpad. Statistical comparison between the two groups was performed using unpaired two-tailed t-test.

The updated Results to describe Figure 4e on page 16, lines 346 to 354 now reads 'To directly assess virus-specific T cell responses in the therapeutic setting, we performed ELISpot analysis on CD4⁺ T cells from the joint footpad at 6 dpi. Consistent with earlier findings, sialokinin-treated mice exhibited a significant reduction in CHIKV-specific IFN- γ -producing CD4⁺ T cells (Fig. 4e), reinforcing its role in dampening local antiviral T cell responses. This reduction in T cell activation was accompanied by lower levels of pro-inflammatory cytokines, including IL-2, IL-27, IL-15, and IL-28, in the joint footpad tissue of sialokinin-treated mice (Fig. 4f), further supporting the anti-inflammatory effect of sialokinin during CHIKV infection'.

Fig. 4: e CD4⁺ T cells were isolated from right joint footpads at 6 dpi (CHIKV, n = 5; CHIKV + sialokinin, n = 5) for CHIKV stimulation and ELISpot assay. Representative images of ELISpot wells show the number of IFN- γ -producing cells in enriched CD4⁺ T cells from joint footpads at 6 dpi. Quantitative plots show the total numbers of IFN- γ -producing CD4⁺ T cells per infected joint footpad. Statistical comparison between the two groups was performed using unpaired two-tailed t-test.

6. It has been previously reported that sialokinin induces vascular leak. The authors in figure 3b show a reduction in CD169⁺ monocytes in the blood of infected mice. What happens to the numbers of other cell populations within the blood?

Response: We appreciate the reviewer's point. In addition to profiling monocytes, we also evaluated neutrophils, as sialokinin has previously been reported to enhance vascular leakage, which could influence neutrophil trafficking. To assess this, we measured circulating neutrophil levels in sialokinin-treated mice during CHIKV infection. When sialokinin was administered during infection, we observed a trend toward higher circulating neutrophil counts in treated mice compared to untreated controls at 3 days post-infection (dpi), however, this difference did not reach statistical significance (Extended Data Fig. 9a). We have updated the Results section on page 12, lines 269 to 275 to read 'Given that sialokinin has been reported to enhance vascular leakage, we also measured circulating neutrophil levels to assess potential changes in vascular permeability and immune cell trafficking. At 3 days post-infection (dpi), we observed a trend toward higher circulating neutrophil counts in sialokinin-treated mice compared to untreated controls. However, this difference did not reach statistical significance (Extended Data Fig. 9a)'.

Extended Data Fig. 9: Effect of sialokinin treatment in CHIKV-infected mice. a, C57BL/6 wild-type (WT) 4-week-old mice (CHIKV, n = 8; CHIKV + sialokinin, n = 10) were infected with 1×10^6 PFU of CHIKV via subcutaneous injection into the joint footpad. For the sialokinin-treated group, mice were infected with CHIKV and simultaneously administered sialokinin (1 μ g). The number of CD11b⁺Ly6G⁺ neutrophils in the blood of CHIKV-infected mice was monitored daily by flow cytometry analysis of whole blood. Data were collated from two independent experiments and are presented as mean \pm SEM. Statistical analysis was performed using the Mann–Whitney U test

We also evaluated the potential impact of therapeutic sialokinin administration by assessing neutrophil levels following intraperitoneal (IP) injection of sialokinin. No significant differences in circulating neutrophil counts were observed between treated and untreated mice at 4 and 6 dpi (Fig. 4b), suggesting that systemic administration of sialokinin does not induce overt vascular leakage. We have updated the Results section on page 15, lines 325 to 332 to read ‘As sialokinin has been reported to enhance vascular leakage, we were concerned that systemic administration via intraperitoneal (IP) injection might exacerbate this effect. To address this, we measured circulating neutrophil levels as an indicator of potential changes in vascular permeability and immune cell trafficking. However, no significant difference in neutrophil counts was observed between sialokinin-treated and untreated mice (Fig. 4b), suggesting that sialokinin administration does not induce significant systemic vascular leakage.

b

Fig. 4: b, Numbers of CD11b⁺Ly6G⁺ neutrophils and CD11b⁺ Ly6C⁺ Siglec-1/CD169⁺ monocytes in the circulation of CHIKV-infected mice, monitored by flow cytometry analysis of whole blood. Data are presented as mean \pm SD (n = 5 per group). Statistical comparisons between the two groups were performed using unpaired two-tailed t-test.

7. The data in the extended data figure 5C and D should be shown in the main figure (Fig. 3)

Response: We appreciate the reviewer’s suggestion. In response, we now have incorporated the key findings from the previous Extended Data Fig. 5D into the main manuscript as part of Fig. 4d, ensuring that the most relevant and significant data are prominently presented.

8. In figure 1d, for the panel showing the quantification of P-PI3K/PI3K and P-Akt/Akt please clarify what time point this represents (20, 40, or 60 minutes post stimulation). Also please clarify what is being compared in the statistical analysis for this figure.

Response: We thank the reviewer for highlighting this. The quantification of P-PI3K/PI3K and P-Akt/Akt in Figure 1d corresponds to the 20-minute time point post-sialokinin stimulation. In the statistical analysis, sialokinin-treated samples were compared to untreated controls to determine the significance of the induced phosphorylation at the 20-minute time point. This comparison was performed using paired two-tailed t-test. More details are now included in the figure legend. The revised legend reads ‘Fig 1: d, Western blot analysis shows that sialokinin stimulation induces activation of the PI3K-Akt signaling pathway in human monocytes. Cells were stimulated with sialokinin for 20, 40 or 60 minutes, resulting in increased levels of phosphorylated PI3K (P-PI3K) and Akt (P-Akt) compared to untreated controls. Pre-treatment with either the NK1R inhibitor CP-96345 or the NK2R inhibitor GR-159897 led to inhibition of PI3K and Akt as early as 20 min post-stimulation. Relative expression levels of P-PI3K/PI3K (n = 4) and P-Akt/Akt (n = 5) in monocytes at the 20-minute time point were quantified with ImageJ. Levels are expressed as fold change relative to the non-treated controls. Bar graphs

represent the mean \pm standard deviation (SD). Statistical comparisons between sialokinin treated sample and untreated control were performed using paired two-tailed t-test. * $p < 0.05$.'

Reviewer #2 (Remarks to the Author):

The manuscript of Fong and co-authors explores the role of a mosquito salivary peptide, sialokinin, in modulating host immune responses to chikungunya virus (CHIKV) infection. The study finds that sialokinin binds to human neurokinin receptors NK1R and NK2R and suppresses activation of monocytes and macrophages, which are key drivers of CHIKV-induced inflammation. Authors show that this immunomodulatory effect is mediated via the PI3K/Akt signaling pathway, leading to reduced expression of inflammatory markers like and diminished production of IFN- γ by CD4⁺ T cells. In mouse models, sialokinin treatment reduced joint swelling, viremia, and pro-inflammatory cytokine levels. Authors speculate that sialokinin could be used as a therapeutic target to mitigate viral-induced inflammation in CHIKV and potentially other mosquito-borne diseases. This overall concept of this study is compelling, however it contains major flaws in statistical analyses and data interpretation, which make me doubt the conclusions of the authors. My major concerns are outlined below.

Response: We thank the reviewer for their thoughtful and constructive feedback, and we appreciate their recognition of the novelty and significance of our study. We acknowledge the reviewer's concerns regarding statistical analyses and data interpretation. In response, we have carefully reviewed our analyses, clarified our rationale where needed. We made appropriate revisions to the manuscript and figure legends to improve transparency and rigor. We address each of the specific concerns in detail below.

1. The significance and fold-change for the differentially expressed genes in Fig 1A is unclear as the volcano plot lacks scale. In addition, authors observe down-regulation of the IFN-stimulated genes (e.g. IFIT2) in sialokinin-treated uninfected cells (line 159). However, ISGs are normally not expressed or expressed at marginal levels. Authors should provide a supplementary table with normalized and raw read counts for all DEGs in this analysis to show whether the expression levels and changes observed are biologically meaningful. Authors should repeat this experiment with CHIKV-infected monocytes to show whether decreased activation is observed in the relevant infection context.

Response: We thank the reviewer for this constructive feedback regarding the transcriptomic analysis presented in Figure 1a. To improve clarity, we have updated the volcano plot in Figure 1a to include appropriate axis labels and scale bars, clearly indicating both the fold-change and statistical significance of differentially expressed genes (DEGs). Additionally, we have provided a Supplementary Table containing both normalized expression values and raw read counts for all DEGs identified in this analysis. This allows for a more transparent evaluation of the biological relevance of the observed changes, including the baseline expression levels of interferon-stimulated genes (ISGs) such as *IFIT2*. The RNA-seq dataset generated for this study has also been deposited in the NCBI Gene Expression Omnibus (GEO) under accession number GSE291153 to ensure transparency and reproducibility.

Fig. 1: Sialokinin modulates monocyte activation via NK receptors and the PI3K/Akt signaling pathway. a, RNA-Seq transcriptional profiling of primary human monocytes (n = 3). The cells were harvested at 24 hours post-treatment for analysis. Volcano plot indicates differentially expressed genes (DEGs) between sialokinin-treated vs. non-treated monocytes. The significantly upregulated genes are highlighted red, while the significantly downregulated genes are highlighted blue. Thresholds for differential gene expression, $|\text{Fold change}| > 0$ and $p < 0.05$, are indicated on \log_2 and \log_{10} scales, respectively.

We acknowledge the reviewer's point that ISGs are typically expressed at low levels in unstimulated cells. However, our analysis revealed a consistent downregulation of several ISGs in sialokinin-treated monocytes, suggesting that sialokinin may suppress basal antiviral readiness even in the absence of infection. To further validate the biological relevance of this observation, we performed an additional gene expression study using CHIKV-infected monocytes, with and without sialokinin treatment. This analysis confirmed that sialokinin treatment leads to downregulation of multiple ISGs, including *Siglec1*, *IFIT2* and *IFI44L*, in the context of active viral infection. These findings are consistent with our phenotypic data showing reduced monocyte activation and support the conclusion that sialokinin dampens antiviral gene expression both at baseline and during infection. We have included the new data in the Extended Data Fig. 7 and also updated the Results section accordingly. The new section in the Results on page 11, lines 241 to 247 now reads 'Total RNA was extracted from harvested monocytes and analyzed by quantitative real-time PCR. Expression levels of key interferon-stimulated genes (ISGs), including *Siglec1*, *IFIT2*, and *IFI44L*, were measured and normalized to housekeeping gene expression using the $\Delta\Delta\text{Ct}$ method. Notably, sialokinin treatment significantly suppressed the expression of all three ISGs not only in non-infected monocytes but also under CHIKV-infected conditions (Extended Data Fig. 7).

Extended Data Fig. 7: Sialokinin treatment attenuates CHIKV infection and suppresses IFN-stimulated gene expression in human monocytes at 24 hours post-infection. Primary human monocytes isolated from six healthy donors were treated with sialokinin (10 μM) and infected with chikungunya virus (CHIKV) at a multiplicity of infection (MOI) of 10. At 24 hours post-infection, intracellular CHIKV levels were quantified by flow cytometry. Total RNA was extracted from harvested cells, and quantitative real-time PCR was performed to assess mRNA expression of interferon-stimulated genes (ISGs), including *Siglec1*, *IFIT2*, and *IFI44L*. Relative gene expression was calculated using the $\Delta\Delta\text{Ct}$ method, normalized to housekeeping gene, and expressed relative to the uninfected control group. Statistical significance was determined using a paired two-tailed t-test ($p < 0.05$, $p < 0.01$).

2. How many replicates were used to calculate Pearson correlation coefficient in Fig 1c? These coefficients are not informative for small sample sizes.

Response: The Pearson correlation matrix in Fig. 1c was calculated using gene expression data from RNA-Seq of primary human monocytes (n = 3). We have now clearly stated the sample size (n = 3) in the figure legends for clarity. While we acknowledge the limitations of correlation analyzes with small sample sizes, the purpose of this analysis was to identify patterns of co-expression among the 12 significant DEGs. This exploratory approach helped to highlight a group of genes enriched in the PI3K-Akt signaling pathway. We subsequently validated the relevance of this pathway using targeted experiments, including Western blot analysis of pathway activation and inhibitor assays, as described in the following sections of the manuscript.

3. Patterns of data points in the groups shown in Fig 1d, 1f, Fig 2, Fig 4c,d, don't follow normal distribution. Therefore, non-parametric statistical tests must be used instead of t-test, as authors correctly did for Fig 5b.

Response: We thank the reviewer for this important observation. In response, we have carefully reviewed our statistical analyzes across all relevant figures. Where data distributions did not meet the assumptions of normality, we have clarified our rationale for the statistical tests used and revised the analyses accordingly. Specifically, we have replaced parametric tests with appropriate non-parametric alternatives where necessary. These updates are reflected in the revised manuscript and figure legends to improve transparency.

4. Data normalization and statistical analyses in Fig 1a and 2 are conceptually flawed as all treated samples have been normalized against control and then compared to it (with the control group having variance of zero). This highly distorts the data unless all samples in the control group had exactly the same value. In addition, it mace data from different groups dependent, while all statistical tests assume independence of data points.

Response: We wish to clarify that the RNA-Seq data were pre-processed using the EPI2ME analysis pipeline, and differential gene expression (DEG) analysis was performed with edgeR to identify genes differentially expressed between sialokinin-treated and untreated monocytes. Our initial intention in Fig. 2 was to illustrate the relative fold differences following sialokinin treatment. However, we acknowledge that normalizing all treated samples to the control group and then statistically comparing them to that same control group could introduce conceptual flaws. In response, we have now revised the graphs to display raw data values and reanalyzed the results using appropriate statistical tests that account for data distribution and independence. We have also updated the figure legends to include a more detailed description of the statistical methods used to improve the clarity of our analysis.

Fig. 2: Sialokinin reduces activation and CHIKV infection in primary human myeloid cells. Human monocytes and monocyte-derived macrophages (MDMs) from ten healthy donors were infected with CHIKV in the presence or absence of sialokinin (10 μ M). **a**, Activation profiles of monocytes and MDMs were assessed by flow cytometry, focusing on CD169 expression. Sialokinin treatment significantly reduced CD169 expression was lower in both monocytes and MDMs following CHIKV infection (MOI = 10) at 24 (monocytes) or 48 (MDMs) hours post-infection. **b**, CHIKV infection was determined using FACS analysis and qRT-PCR on the viral supernatant. Viral load was quantified by qRT-PCR targeting CHIKV nsP1 viral copies. Data were \log_{10} -transformed prior to statistical analysis. CHIKV infectivity toward monocytes and MDMs was reduced in the presence of sialokinin. Statistical analyses were performed using either a paired two-tailed t-test or a Wilcoxon signed-rank test, depending on data distribution. $p < 0.05$, $p < 0.01$ indicate statistically significant differences compared to untreated controls.

5. Considering that virus replicates logarithmically, data on viral RNA copies in Fig 2b must be \log_{10} transformed prior to statistical comparison. Values must be plotted on logarithmic scale. What is the reason for not showing individual data point on viral copy number plots. What were the viral titers in the culture fluids? Additionally, the selection of the time point for this analysis is not justified. Authors should determine viral growth kinetics through multiple time points.

Response: We thank the reviewer for this feedback. In response, we have revised the data presentation in Fig. 2b by applying a \log_{10} transformation to the viral RNA copy numbers prior to statistical analysis. The updated figure now displays the data on a logarithmic scale. We also wish to clarify that the viral RNA measurements were obtained from the culture supernatant. Individual data points have now been included in the revised plot to enhance transparency and allow better assessment of data variability. We have also revised the figure legend to improve the clarity.

To clarify, we did perform kinetic experiments at 24- and 48-hours post-infection (hpi) to determine the optimal time points for assessing CHIKV infectivity and CD169 expression on both human monocytes and monocyte-derived macrophages (MDMs). Our results indicate that 24 hpi is optimal for monocytes, while 48 hpi is optimal for MDMs. These time points were selected based on the observed differences in both infectivity and CD169 expression. We have now included these data in the Extended Data Figures 5 & 6 and updated the Results section on page 11, lines 225 to 232 to read 'To determine the optimal time points for assessing CHIKV infectivity and CD169 expression following sialokinin treatment, kinetic experiments were performed at 24- and 48-hours post-infection (hpi) using both human monocytes and monocyte-derived macrophages (MDMs). These experiments revealed that 24 hpi is optimal for monocytes, while 48 hpi is optimal for MDMs, based on the observed differences in magnitude of infection and CD169 expression levels with sialokinin treatment (Extended Data Figs. 5 & 6). These time points were subsequently used in all downstream analyses'.

Extended Data Fig. 5: Effect of sialokinin on human monocyte activation and CHIKV infection at 24- and 48-hours post-infection. Human monocytes from healthy donor were infected with CHIKV in the presence or absence of sialokinin (10 μ M). Activation profiles of monocytes were assessed by flow cytometry. CD169 expression was lower in sialokinin-treated cells following CHIKV infection (MOI = 10) at 24 hours post-infection. CHIKV infectivity in monocytes was reduced in the presence of sialokinin.

Extended Data Fig. 6: Effect of sialokinin on human monocyte-derived macrophages (MDMs) activation and CHIKV infection at 24 and 48 hours post-infection. Human MDMs from healthy donor were infected with CHIKV in the presence or absence of sialokinin (10 µM). Activation profiles of monocytes were assessed by flow cytometry. CD169 expression was lower in sialokinin-treated cells following CHIKV infection (MOI = 10) at 24 hours post-infection. CHIKV infectivity in MDMs was reduced in the presence of sialokinin.

6. Data in Fig3C and D is conflicting with 4B, as on one hand treatment reduces viral load and inflammation while in figure 4B there is no effect on viral load and mild effects on inflammation. This suggests the sialokinin may be affecting viral replication and the reduced inflammation may simply result from reduced viral load. This should be discussed in the paper.

Response: We thank the reviewer for this thoughtful comment. To address the apparent discrepancy between Fig. 3c,d and Fig. 4b, we have performed additional experiments to better characterize the temporal and immunological effects of sialokinin.

Importantly, the sialokinin treatment in Fig. 4b was introduced at 2 dpi, which is one day before the peak of viremia. At this stage, virus replication is already well established, potentially limiting the observable impact of sialokinin on viral load. In contrast, earlier exposure to sialokinin (as in Fig. 3) may facilitate virus dissemination and immune modulation before peak viremia, leading to more pronounced effects on both viral load and inflammation.

To further investigate this, we quantified tissue viral loads and performed virus-specific CD4⁺ T cell ELISpot assays. These revealed that sialokinin enhances early virus dissemination and suppresses virus-specific T cell responses, likely through early inhibition of monocyte activation and antigen presentation. We have revised the manuscript to discuss the multifaceted role of sialokinin in CHIKV pathogenesis.

The revised Discussion on page 19, lines 403 to 425 reads:

'Early in infection, we observed increased viral load in the contralateral (left) footpad at 1 dpi in sialokinin-treated mice, indicating that sialokinin facilitates systemic virus dissemination. This supports the hypothesis that mosquito-derived sialokinin promotes virus spread beyond the inoculation site via vascular leakage. Interestingly, despite this early enhancement of virus dissemination, viral titers in the inoculated footpad were comparable between groups by 6 dpi. However, we detected a significant reduction in virus-specific CD4⁺ T cell responses in sialokinin-treated mice. This apparent discrepancy likely reflects the temporal dynamics between virus replication and immune activation. While virus replication may have plateaued or declined by 6 dpi, early sialokinin-driven modulation of monocytes had already shaped the downstream T cell response.'

We propose that sialokinin exerts its immunomodulatory effects by suppressing monocyte activation during the early phase of infection. Monocytes are key antigen-presenting cells, and their activation status directly influences cytokine production and T cell priming. In our study, sialokinin treatment led to a marked reduction in monocyte and macrophage activation, as evidenced by decreased Siglec-1/CD169 expression. Importantly, we also observed a reduction in circulating activated monocytes, suggesting that sialokinin's immunosuppressive effects extend beyond the local tissue environment to influence systemic immune responses. This early suppression likely impairs antigen presentation and cytokine signaling, resulting in diminished virus-specific CD4⁺ T cell priming and differentiation.

We also conclude the multifaceted roles of sialokinin on page 22, lines 481 to 486. The section reads:

'Taken together, our findings underscore the multifaceted role of sialokinin in CHIKV pathogenesis. It facilitates early virus dissemination, suppresses monocyte activation both locally and systemically, and impairs virus-specific CD4⁺ T cell responses. These insights expand upon previous work and highlight sialokinin's potential as a therapeutic target for modulating inflammation in CHIKV and other immune-mediated diseases'.

Reviewer #3 (Remarks to the Author):

The manuscript by Fong et al. explores the immunomodulatory effects of the *Aedes aegypti* salivary peptide sialokinin during Chikungunya virus (CHIKV) infection. It is a novel and impactful study where the authors demonstrate that sialokinin binds to neurokinin receptors on monocytes and limits their activation, evidenced by downregulation of several type-I interferon stimulated genes (ISGs) and surface activation markers, as well as their infectivity. The study links these findings to downstream inflammation and T cell activation both in vitro and in vivo, and further correlates anti-sialokinin IgG titers with disease severity and inflammation in CHIKV patients.

The manuscript is well executed and addresses an important aspect in our understanding of vector-host-virus immunological dynamics. The integration of transcriptomics, functional assays, in vivo models and patient samples is advantageous and exciting. The mechanistic insights into neurokinin receptor signaling are novel and presented with clarity and overall, the findings are robust and of potential translational relevance.

Response: We sincerely thank the reviewer for the positive and encouraging feedback. We are pleased that the novelty, mechanistic insights, and translational potential of our study were appreciated, as well as the integration of diverse approaches including transcriptomics, functional assays, *in vivo* models, and patient data. We are grateful for your recognition of the robustness and clarity of our findings. We have taken care to further strengthen the manuscript by addressing all additional comments in detail.

Below are minor comments:

- The authors cite prior work (Reference 32) showing that mosquito saliva enhances infection through sialokinin-dependent vascular leakage. A brief discussion of how their findings reconcile with this study would be helpful and could contextualize the multifaceted role of sialokinin during CHIKV infection.

Response: We thank the reviewer for highlighting the relevance of prior work demonstrating that mosquito saliva enhances virus infection through sialokinin-dependent vascular leakage. Our findings build upon and extend this work by elucidating the multifaceted role of sialokinin during CHIKV infection.

While the previous study emphasized sialokinin's role in promoting early viral dissemination via vascular leakage, our data reveal additional immunomodulatory effects. Specifically, we show that sialokinin suppresses monocyte activation and impairs virus-specific CD4⁺ T cell responses, suggesting that its impact extends beyond facilitating virus spread to modulating host immunity.

We have incorporated a section in the revised discussion to reconcile our findings with prior work and to contextualize the broader role of sialokinin in CHIKV pathogenesis.

The revised Discussion on page 19, lines 403 to 425 reads:

'Early in infection, we observed increased viral load in the contralateral (left) footpad at 1 dpi in sialokinin-treated mice, indicating that sialokinin facilitates systemic virus dissemination. This supports the hypothesis that mosquito-derived sialokinin promotes virus spread beyond the inoculation site via vascular leakage. Interestingly, despite this early enhancement of virus dissemination, viral titers in the inoculated footpad were comparable between groups by 6 dpi. However, we detected a significant reduction in virus-specific CD4⁺ T cell responses in sialokinin-treated mice. This apparent discrepancy likely reflects the temporal dynamics between virus replication and immune activation. While virus replication may have plateaued

or declined by 6 dpi, early sialokinin-driven modulation of monocytes had already shaped the downstream T cell response.

We propose that sialokinin exerts its immunomodulatory effects by suppressing monocyte activation during the early phase of infection. Monocytes are key antigen-presenting cells, and their activation status directly influences cytokine production and T cell priming. In our study, sialokinin treatment led to a marked reduction in monocyte and macrophage activation, as evidenced by decreased Siglec-1/CD169 expression. Importantly, we also observed a reduction in circulating activated monocytes, suggesting that sialokinin's immunosuppressive effects extend beyond the local tissue environment to influence systemic immune responses. This early suppression likely impairs antigen presentation and cytokine signaling, resulting in diminished virus-specific CD4⁺ T cell priming and differentiation.

We also conclude the multifaceted roles of sialokinin on page 22, lines 481 to 486. The section reads:

'Taken together, our findings underscore the multifaceted role of sialokinin in CHIKV pathogenesis. It facilitates early virus dissemination, suppresses monocyte activation both locally and systemically, and impairs virus-specific CD4⁺ T cell responses. These insights expand upon previous work and highlight sialokinin's potential as a therapeutic target for modulating inflammation in CHIKV and other immune-mediated diseases'.

- The authors should consider measuring phosphorylation of PI3K and AKT in PBMCs derived from mice treated with sialokinin to strengthen the hypothesis that this is the relevant signaling pathway activated.

Response: We appreciate the reviewer's suggestion. Neurokinin receptors are known to be highly conserved across species¹. To further substantiate our hypothesis, we conducted additional experiments using the murine macrophage cell line RAW 264.7. In these cells, sialokinin's ability to suppress monocyte activation, evidenced by reduced CD169 expression, was abrogated upon treatment with a PI3K signaling inhibitor. These results support the involvement of PI3K signaling in mediating the immunomodulatory effects of sialokinin in mice.

Extended Data Fig. Sialokinin suppresses macrophage activation via a PI3K-dependent pathway in murine macrophages. RAW 264.7 murine macrophage cells were treated with sialokinin in the presence or absence of a PI3K signaling inhibitor. Macrophage activation was assessed by measuring surface CD169 expression. Sialokinin reduced CD169 expression, indicative of suppressed activation. This effect was abrogated when PI3K signaling was inhibited, suggesting that sialokinin mediates its immunomodulatory effect through the PI3K pathway in murine macrophage. Data are representative of two independent experiments. ****p* < 0.001 by one-way ANOVA followed by post-hoc Tukey's multiple comparison tests

1. Mishra, A., & Lal, G. (2021). Neurokinin receptors and their implications in various autoimmune diseases. *Current Research in Immunology*, 2, 66–78. <https://doi.org/10.1016/j.crimmu.2021.06.001>

- The human patient cohort seems to have a sex bias (Out of 30 individuals, 26 are male vs. 4 female). The authors should discuss how this may impact interpretation.

Response: We thank the reviewer for this important observation. We acknowledge that our human cohort has a male predominance (26 males and 4 females), which may influence immune responses and disease outcomes, as sex-based differences in immunity are well-documented. In response to this comment, we have included a statement in the Discussion (lines 498 to 503), which reads 'Notably, our human patient cohort had a sex imbalance (26 males vs. 4 females), which may influence immune responses and disease severity. While our key findings remain robust, future studies with more balanced cohorts are needed to assess potential sex-specific differences in antibody responses to mosquito saliva components and their impact on disease outcomes'. This addresses the limitation and highlights the importance of future studies to further investigate potential sex-specific differences in immune responses to mosquito saliva and their contribution to disease severity.

- In line 60, the sentence "in a warming sicker world" is impactful but rephrasing it could be considered.

Response: We thank the reviewer for the helpful suggestion. We have revised the sentence to read 'Given the growing threat of mosquito-borne diseases in a warming, disease-burdened world, modulating the action of mosquito salivary factors like sialokinin could offer a novel therapeutic strategy to mitigate viral-induced inflammation and improve clinical outcomes'.

- In the in vitro experiments (Figures 1 and 2) it is not clear for how long the incubation with sialokinin was.

- The figure legends should include an indication how many experiments were conducted.

Response: Detailed information on the n number and experimental conditions is now included in the updated figure legends.

Reviewer #4 (Remarks to the Author):

Response: We thank you for your contribution to the peer review process and appreciate the thoughtful feedback provided.

Reviewer #5 (Remarks to the Author):

Response: We thank you for your contribution to the peer review process and appreciate the thoughtful feedback provided.